# The impact of human dispersals and local interactions on the genetic diversity of coastal Papua New Guinea over the past 2,500 years

The inhabitants of New Guinea and its outlying islands have played an important role in the human history of the Pacific region. Nevertheless, the genetic diversity, particularly of pre-colonial communities, is still understudied. Here we present the ancient genomes of 42 individuals from Papua New Guinea (PNG). The ancient genomic results of individuals from Watom Island (Bismarck Archipelago) and the south and northeastern coasts of PNG are contextualized with new (bio-) archaeological data. The individuals' accelerator mass spectrometry (AMS) dates span 2,500 years of human habitation, and our results demonstrate the influences of different dispersal events on the genetic make-up of ancient PNG communities. The oldest individuals show an unadmixed Papuan-related genetic signature, whereas individuals dating from 2,100 years before present carry varying degrees of an East-Asian-related contribution. These results and the inferred admixture dates suggest a centuries-long delay in genetic mixture with local communities after the arrival of populations with Asian ancestry. Two geographically close communities on the South Coast, AMS dated to within the past 540 years, diverge in their genetic profiles, suggesting differences in their interaction spheres involving groups with distinct ancestry. The inferred split time of these communities around 650 years before present coincides with intensified settlement activity and the emergence of regional trade networks.

Humans occupied Near Oceania from at least 50,000 years before present (BP)[1–5]. The region comprises the Aru Islands, New Guinea, the Bismarck Archipelago and the Solomon Islands. Despite the critical role that its occupants have played in the genetic history of the adjacent regions of Indonesia[6,7], northwestern Remote Oceania[8] and western Remote Oceania[9–12], its past genetic diversity remains unstudied through ancient genomes.

Apart from being the setting of some of the earliest maritime dispersals by modern humans, Near Oceania was the embarkation point for human dispersals into Remote Oceania, culminating with the settlement of some of the last islands on earth to be permanently inhabited.

As part of a dispersal throughout island Southeast Asia, starting around 5,000 BP[13–15], people, presumably Austronesian-speaking, arrived in the Bismarck Archipelago by 3,300–3,200 BP[16,17]. Recognized in the archaeological record by distinct ornately decorated pottery and a lifestyle that incorporated horticulture, domesticated animals, long-distance seafaring and maritime and terrestrial foraging[18,19], this combination of traits is known as the Lapita cultural complex[20]. The linguistic diversity of modern Near Oceania includes closely related languages of the Austronesian family and various unrelated non-Austronesian (that is, Papuan) languages, thought to have derived from languages spoken by the region's original inhabitants[21]. Archaeological evidence suggests

✉e-mail: kathrin_naegele@eva.mpg.de; contact@bioarchsouth.com; cosimo.posth@uni-tuebingen.de; krause@eva.mpg.de

this diversity is the result of high mobility, a deeply complex history of interactions before the arrival of Austronesian-speaking peoples, their repeated settlement of the coasts[22–24] and interactions with local inhabitants. However, the nature and full extent of the interactions remain unknown[25].

Until recently, occupation sites of Lapita-associated populations in Near Oceania seemed primarily restricted to smaller offshore islands near the larger island arcs[26]. Archaeological excavations have challenged this view for the middle and late Lapita period on mainland Papua New Guinea (PNG) with the discovery of sites featuring Lapita pottery and associated settlements in well-dated contexts on the south coast of PNG[27] and the Massim region off of southeastern New Guinea[28]. A small number of Lapita sites are documented on the north coast of New Guinea and the Vitiaz Strait. The Bismarck Archipelago is home to a small number of Lapita sites, suggesting similar timing and process of settlement to the south coast, and implying that Austronesian languages appeared or proliferated in these regions during the middle to late Lapita period[29–33].

From 2,200 BP onwards, intensive settlements on the south coast of PNG are associated with shell-impressed pottery and tools manufactured in an exchange sphere extending across ~700 km (refs. 25,34), providing evidence for an occupation by groups culturally and perhaps genetically descended from the Lapita Cultural Complex[27,35,36]. The restricted present distribution of Austronesian-speaking people along the south coast[37] supports this idea, as do the local oral traditions[38,39]. In the period between 1,200 and 500 BP, a major change in settlement strategies is observed, and long-distance connections between coastal communities were disrupted. During this so called 'Papuan Hiccup'[40] or 'Ceramic Hiccup'[41], settlement along the eastern south coast continued, but locations further west, previously inhabited for centuries, were abandoned, suggesting the re-organization and relocation of populations locally and regionally[42]. Pottery with increasingly localized forms and decorative motifs emerged within this timeframe[40–43]. After 700 BP, many new settlements were established, and several former locals were reoccupied. However, it remains unclear whether this is the return of descendants of the previous occupants or the arrival of new populations[44].

This study provides a genetic perspective on the diversity of ancient inhabitants of different regions in PNG and the Bismarck Archipelago. We investigate critical questions regarding the different ancestries involved in the formation of the genetic make-up in the region, specifically the genetic impact of the dispersal of Austronesian-speaking groups. We provide further insights on different human dispersal events or migration processes in the region, such as the settlement of Vanuatu and the Mariana Islands. The genetic data are contextualized with partly new (bio-) archaeological data in the form of isotopic information and microparticles from dental calculus to place these individuals into a cultural and environmental context in PNG by providing evidence for patterns of human diet and mobility.

## Results

### AMS dating, corrections, isotopic and microparticle evidence

To understand possible chronological changes in ancestry, we produced new accelerator mass spectrometry (AMS) radiocarbon dates for 28 individuals and included previously published dates for four individuals (Supplementary Tables 1 and 2). Stable isotope data were used to correct for the marine reservoir effect[45–49], observed as a result of marine based diets, when necessary[50–54].

The radiocarbon determinations (Supplementary Table 2) date the individuals at Nebira 2 on the southern coast from 540–320 cal BP (calibrated years before present) up to 420–0 cal BP. Largely overlapping in time, individuals from the nearby site of Eriama date to around 470–310 cal BP to sometime within the last 280 cal BP. An individual from the site of Tilu, on the northeastern coast, dates to

around 690–500 cal BP. The five individuals excavated from Watom show wide time intervals, covering a period from 2,690–2,110 cal BP to 630–510 cal BP (Table 1 and Fig. 1b). From the stable isotope data, Eriama and Nebira 2 were found to be consuming a fully terrestrial diet. Tilu and Watom individuals consumed a partially marine diet, consistent with zooarchaeological analysis[55,56], and required AMS dating corrections (Supplementary Tables 2 and 6).

Additionally, the isotope data provided information about past mobility and diet at the sites (Supplementary Table 6 and Extended Data Fig. 1). For Watom in the Bismarck Archipelago, human strontium isotope ($^{87}Sr/^{86}Sr$; Supplementary Table 4 and Extended Data Fig. 1c) data were consistent with a local signature for all individuals, suggesting an upbringing on the island or one with a similar underlying geology. On the south coast of PNG, wallaby ($n = 13$) tooth enamel $^{87}Sr/^{86}Sr$ ratios provide a local baseline, helping to interpret local and non-local individuals at the Nebira 2 site (Supplementary Tables 5 and 6 and Extended Data Fig. 1d). The majority of human $^{87}Sr/^{86}Sr$ ratios analysed from the Nebira 2 assemblage were consistent with the local signature, suggesting the childhood of these individuals was spent at the site or nearby. Five individuals displayed a non-local $^{87}Sr/^{86}Sr$ ratio, suggesting they spent their early years away from the site, possibly nearer to the coast, before moving to the site sometime before their death[54]. The results of a Ward's hierarchical cluster analysis of the $\delta^{13}C_{collagen}$, $\delta^{15}N_{collagen}$, $\delta^{13}C_{dentine}$, $\delta^{15}N_{dentine}$, $\delta^{13}C_{carbonate}$ and $^{87}Sr/^{86}Sr$ of the humans from the Nebira site (Extended Data Fig. 1e) show that the non-local individuals grouped together, indicating their child diet, adult diet and childhood residency were similar to one another but different than the 'local' Nebira individuals. The tooth $\delta^{13}C_{carbonate}$ and $\delta^{13}C_{dentine}$ values of these five non-local individuals were higher than the 'local' individuals from the site, indicating that they may have been eating more marine foods at a young age (Supplementary Table 6b). The $\delta^{13}C_{dentine}$, $\delta^{15}N_{dentine}$ and $\delta^{13}C_{enamel}$ values for the wallabies provided a reference for herbivores eating $C_3$ and $C_4$ plants (Extended Data Fig. 1f). The human dietary isotope results suggest that both the Nebira and Watom communities were eating a diet high in terrestrial resources such as horticultural foods and wild and domestic animals that included wallaby at the Nebira site on the south coast (Supplementary Table 2 and Extended Data Fig. 1f).

Additionally, microparticle analysis recovered from the dental calculus at Watom showed that arboriculture was an important part of the diet. The majority of phytoliths recovered originated from palm or other trees (Supplementary Table 7). Some of the tree phytoliths may also indicate medicinal use as they are likely to come from the bark or leaves of trees[57].

### Ancient genetic variation

To explore the genetic variation in ancient Near Oceania and archaeological evidence of mobility in the region, we generated genome-wide data for 41 ancient individuals from the sites on the PNG mainland and the Bismarck Archipelago (Supplementary Table 1 and Supplementary Information). To overcome the poor ancient DNA (aDNA) preservation, we used a targeted capture approach, enriching for 1.2 million single nucleotide polymorphisms (SNPs) across the genome. Contamination estimates (Supplementary Table 9) were low with an average of 2% contamination.

Genetic variability between present day, published ancient and newly produced ancient genomes was investigated through a principal component analysis (PCA) (Supplementary Table 10 and Fig. 2) and an ADMIXTURE analysis (Extended Data Fig. 2). Ancient individuals from the Papuan coast included in this study do not cluster with present-day individuals from the PNG highlands but instead with present-day populations inhabiting the southern and northern coastlines of PNG and the Massim Archipelago. Illustrating a cline, extending from the present-day individuals from the PNG highlands to present-day East Asian and ancient Early Remote Oceanians (ERO), the individuals from

**Table 1 | Overview of the individuals in this study**

| Region | Site | Lab ID | Date (95% cal BP) | Genetic sex | Mt Haplogroup | Y-haplogroup | Coverage1240 |
|---|---|---|---|---|---|---|---|
| Bismarck Archipelago | Reber–Rakival | WAT001 | 670–550 | XX | M29a | – | 650.194 |
| | | WAT002 | 2,050–1,610 | XX | B4a1a1 | – | 239.793 |
| | | WAT003 | 630–510 | XX | M29a | – | 638.382 |
| | | WAT005 | 3,830–3,370* | XY | P1 | C1b2 | 34.473 |
| | | WAT006 | 2,690–2,110 | XY | NA | M1 | 10.810 |
| PNG south coast | Nebira | NBR001 | 530–320 | XX | B4a1a1a | – | 171.477 |
| | | NBR002 | rel of NBR023 | XY | B4a1a1a | C1b2 | 307.064 |
| | | NBR003 | 500–310 | XX | B4a1a*1 | – | 616.322 |
| | | NBR004 | 510–320 | XY | Q1 | M1 | 666.268 |
| | | NBR005 | 480–310 | XY | B4a1a*1 | O2a2b2 | 341.762 |
| | | NBR006 | NA | XX | F1a3a | – | 557.355 |
| | | NBR007 | 420–0 | XY | B4a1a*1 | C1b2 | 403.661 |
| | | NBR008 | NA | XY | B4a1a*1 | – | 11.350 |
| | | NBR009 | 500–320 | XY | B4a1a*1 | C | 365.649 |
| | | NBR010 | 440–150* | XY | B4a1a*1 | M1a1a3b | 552.845 |
| | | NBR011 | NA | XY | B4a1a*1 | C1 | 517.271 |
| | | NBR012 | NA | XY | B4a1a1a | C1b2 | 474.890 |
| | | NBR013 | 480–310/510–310 | XX | Q1 | – | 533.927 |
| | | NBR014 | 540–500 | XY | B4a1a1 | C1b2 | 369.415 |
| | | NBR015 | 530–470 | XY | B4a1a*1 | C1b2a1c | 543.129 |
| | | NBR016 | NA | XX | B4a1a*1 | – | 621.244 |
| | | NBR017 | 500–320 | XY | P1 | C1b2a1c | 337.929 |
| | | NBR018 | 460–310 | XX | B4a1a*1 | – | 619.110 |
| | | NBR019 | 460–310 | XY | B4a1a*1 | C1b2 | 682.573 |
| | | NBR020 | 510–320 | XY | B4a1a*1 | O2a2b2a2b | 355.084 |
| | | NBR021 | 430–150 | XX | B4a1a*1 | – | 500.416 |
| | | NBR022 | NA | XY | NA | NA | 34.955 |
| | | NBR023 | NA | XX | B4a1a*1 | – | 476.202 |
| | | NBR024 | rel of NBR023 NBR012 NBR002 | XY | B4a1a*1 | NA | 305.076 |
| | | NBR025 | NA | XY | B4a1a*1 | M1a1a3b | 634.416 |
| | | NBR026 | NA | XX | B4a1a*1 | – | 234.122 |
| | Eriama | ERI002 | 430–150 | XX | B4a1a*1 | – | 637.175 |
| | | ERI003 | 470–310 | XX | B4a1a*1 | – | 428.181 |
| | | ERI004 | 280–0 | XX | NA | – | 712.904 |
| | | ERI005 | 280–0 | XY | B4a1a*1 | C1b2 | 568.819 |
| | | ERI006 | 470–310 | XY | B4a1a*1 | S1a1b1d2b~ | 673.199 |
| | | ERI007 | 300–0 | XX | NA | – | 673.781 |
| | | ERI008 | 310–0 | XX | B4a1a*1 | – | 667.993 |
| | | ERI009 | 470–310 | XY | B4a1a1a | C1b2 | 789.118 |
| PNG north coast | Tilu | TIL001 | 690–500 | XY | B4a1a1 | S1d | 24.180 |
| | | TIL002 | NA | – | – | – | – |
| | | TIL003 | NA | – | – | – | – |
| | | TIL004 | NA | XY | B4a1a1 | NA | 7.413 |
| | Nunguri | NUN001 | NA | – | NA | NA | – |
| | | NUN002 | NA | – | NA | NA | – |

Provided are geographic region, site, individual ID, date range 95% cal BP, genetic sex, mitochondrial (Mt) haplogroups, y-chromosomal haplogroups, markers covered out of 1.2 million (coverage1240). The asterisk indicates that the date removed due to poor quality (Supplementary Table 2); samples from Nunguri did not yield sufficient DNA. Full information in Supplementary Tables 1 and 2. NA is not available.

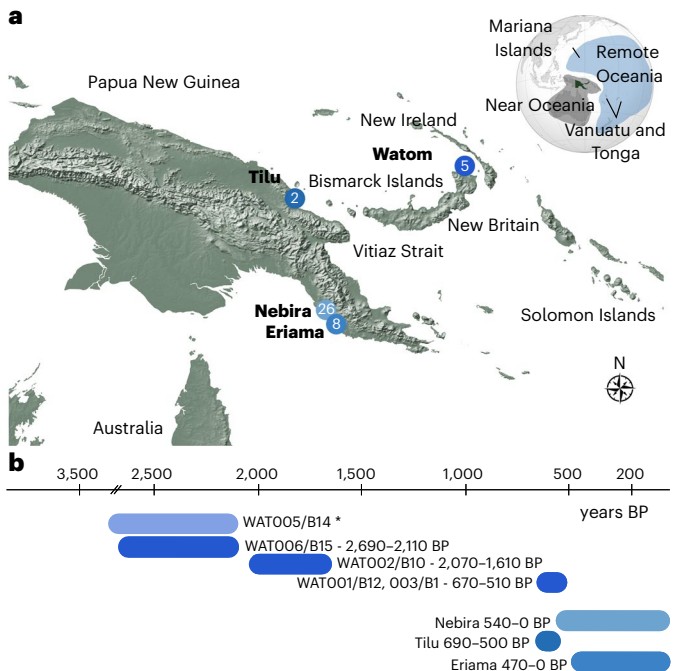

**Fig. 1 | Sites and samples. a**, Map of Near Oceania showing the location of the sites discussed in this study, the number of individuals analysed per site and other places mentioned in the text. **b**, Date ranges for each site/individual in calibrated years BP. Date ranges are based on directly dated skeletal remains and do not necessarily represent the entire occupation of the site. The colours indicate the different archaeological sites or complexes and are used to represent the individuals from these sites in other figures. The asterisk indicates that the date is based on layer information[70].

Eriama on the southern coast cluster on the Papuan end of the cline together with the two individuals from the site of Tilu on the northern coast. However, two individuals (ERIOO4/ACV-4 and ERIOO6/ACV-6) are removed from the group and shifted towards East Asians. Individuals from the nearby site of Nebira 2 cluster close together around the midpoint of the same cline. The observations from the PCA are supported by a genetic grouping analysis using $f_4$-statistics and qpWave (Supplementary Table 11), confirming a genetically homogeneous community at Nebira 2, whereas individuals from Eriama form two groups. The placement of the Eriama and Nebira 2 individuals suggests a genetic composition derived from a mixture of Papuan-related ancestry most similar to present-day highland populations of PNG (Extended Data Fig. 3) and of East-Asian-related ancestry. The three older individuals excavated from Watom were analysed separately (WAT002/B10, WAT005/B14, WAT006/B15), considering low coverage in WAT006/B15. WAT005/B14 and WAT006/B15 were consistent with exclusively Papuan-related ancestry (Supplementary Table 12) and have a considerable time interval of >600 years to WAT002/B10 (1900 BP and WAT005/B14 ~ 2600 BP), which also showed a notable shift towards Asian populations in the PCA (Figs. 1b and 2 and Supplementary Tables 1 and 2). WAT001/B1 and WAT003/B12 were grouped based on more recent dates and genetic similarity.

### Ancestry modelling

Having established that, except for WAT005/B14 and WAT006/B15, none of the individuals derived from only one stream of ancestry using qpWave (Supplementary Table 12), we modelled the East-Asian-related ancestry using present-day Indigenous Taiwanese (Amis) and the Papuan-related ancestry using New Guinea Highlanders. qpAdm tests (Supplementary Table 12) show that 34 of 36 individuals can indeed be modelled as a mixture of these two ancestries (Fig. 3a). However, $f_4$-statistics add detail, showing ancient individuals from the mainland of Papua New Guinea and the Bismarck Archipelago differ in their affinities to present-day Near Oceanic populations. Whereas the individuals from Eriama, Nebira 2 and Tilu show higher affinity to highland New

Guinean populations, all individuals from Watom show higher affinities to Baining-speakers (a non-Austronesian language) from New Britain (Supplementary Table 11 and Extended Data Fig. 3a). The Nebira 2 and Eriama individuals with higher East-Asian-related ancestry proportions show more affinity to the ERO from Vanuatu and Tonga compared to present-day Indigenous Amis from Taiwan, ~2,650-year-old individuals from Guam or ~4,500-year-old individuals from the site of Suogang on Penghu Island, west of Taiwan (Supplementary Table 11 and Extended Data Fig. 3b).

The patterns observed in the PCA and $f_4$-statistics are supported by the qpAdm analysis. East-Asian-related ancestry in individuals from Nebira 2 ranges from 45–60% and is higher compared to Eriama and Tilu, where the proportion is around 20%. The two individuals from Eriama (ERIOO4/ACV-4 and ERIOO6/ACV-6) not included in the main group are the exception, showing higher East-Asian-related ancestry proportions of around 35%.

The individual WAT002/B10, dated to around 1,900 BP, shows admixture between Papuan-related and East-Asian-related ancestry, at 40% to 60%, respectively. However, the East-Asian-related proportion is reduced to 20% when ancient individuals are used as a proxy for this ancestry (Fig. 3c).

### Timing of the admixture events

To estimate the time of admixture between the two ancestry components present in the individuals, we performed an admixture dating analysis using DATES (Fig. 3d and Extended Data Fig. 4). For the ~1,900-year-old individual from Watom Island (WAT002/B10), we inferred an admixture around ten generations before, resulting in a date of ~2.100 BP for the admixture event and for the two individuals dating to ~500 BP from the same site.

We again observe slight differences between the two sites on the southern PNG coast. Inferred from the dated individuals only, individuals from Eriama have an average admixture date of 1,100 BP (ranging from 1,600 to 600 BP), whereas the average date for Nebira 2 is slightly older (around 1,500 BP, ranging from 1,800 to 900 BP) (Supplementary

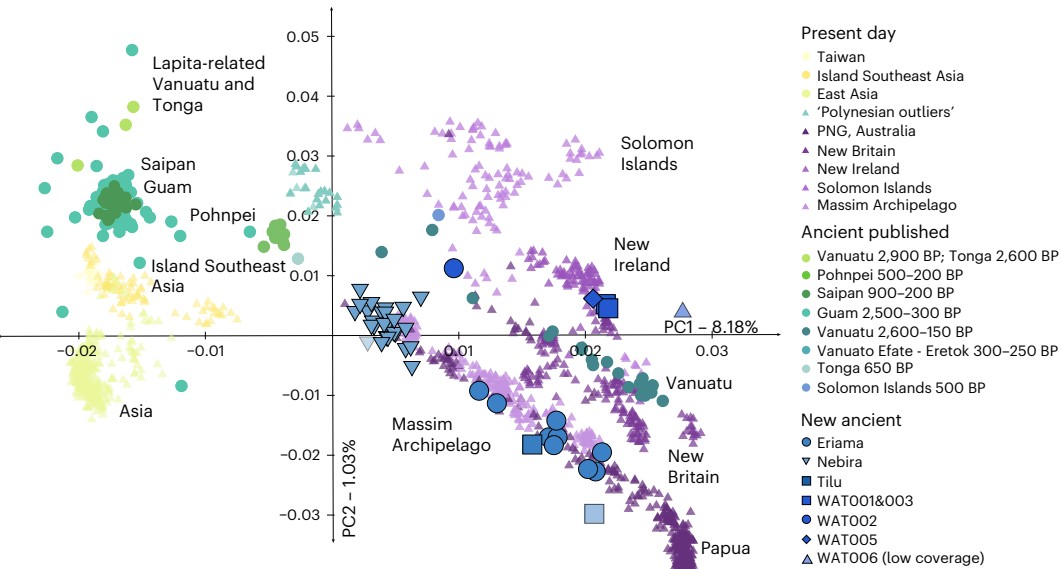

**Fig. 2 | Population substructure in Oceania.** PCA of present-day individuals (upwards triangles) from Asia, Island Southeast Asia (yellow colours, left side of the plot), Near Oceania (purple colours, right side of the plot) and Remote Oceania (green and turquoise colours) with ancient individuals (circles) projected. Outlined individuals are newly reported in this study. Individuals with insufficient number of SNPs (< 20 000) are shown transparent (WAT006, NBR008 and TIL004).

Table 13). ERI007/ACV-7 and NBR020/ACJ-34 are the exception with younger admixture dates of 370 BP and 650 BP, respectively. Coincidently, NBR020/ACJ-34 displayed a non-local $^{87}Sr/^{86}Sr$ ratio and childhood dietary values (Extended Data Fig. 1d). Because of low coverage, individuals from Tilu were grouped for this analysis, resulting in an inferred admixture event of 900 BP.

### Sex-biased mixture

We determined the Y-chromosomal haplogroups for 21 of the 25 genetically male individuals and mitochondrial DNA (mtDNA) haplogroups for 38 of the 41 individuals in the dataset (Table 1 and Extended Data Fig. 5).

The majority of individuals carry mtDNA haplogroups associated with present-day and ancient East Asian populations[58–60], dominated by the mtDNA haplogroup B4a1a1. Only four individuals carry Papuan-related mtDNA haplogroups. Conversely, the Y-chromosome haplogroups indicate the opposite pattern. Apart from one Y-chromosome haplogroup with East Asian origin (O2a2b2), all other 18 identified haplogroups are most common in Near Oceania (Table 1 and Extended Data Fig. 5). The pattern of sex-biased mixture is also detectable at a genome-wide level, shown by the analysis of mixture proportions on the X chromosome, comparing to those inferred from the autosomes. The analysis shows an excess of Austronesian-related ancestry on the X chromosome ranging from 10 to 60 percentage points, suggesting a sex-biased mixture as previously observed in similar to ancient individuals from Vanuatu and Wallacea[7,9] (Supplementary Table 12).

### Settlement sequence of the broader region

To investigate the relationships with other ancient groups within the region, we modelled an admixture graph using qpGraph. The source region for the initial settlement of the Mariana Islands is under debate. Radiocarbon dates indicate settlement ~3,200 BP[61], but some archaeological evidence suggests even earlier occupation several centuries earlier[62]. Potential source regions include the Philippines[63,64] and, based on highest probabilities of landfall in the Marianas during a sea voyage, northern Near Oceania[65]. In the absence of ancient genomes of reasonable quality from the Philippines, we evaluate this indirectly. Using ancient genomes from Guam and Saipan, dating to between 2,800 and

200 BP[8,66], we tested two competing hypotheses. First, the Marianas could have been settled from the Philippines, in which case the group is modelled to have split before the formation of the Lapita-related genetic component, including the East-Asian-related component in WAT002/B10 (Extended Data Fig. 6b). Alternatively, the Marianas could have been settled from Near Oceania, in which case their group should split after they arrived in the Bismarck Archipelago (Extended Data Fig. 6c). The data provide a better fit for the model of a split of the genomes from Guam and Saipan before WAT002/B10 (|$Z$| = 2.883 vs |$Z$| = 4.196 for the alternative settlement from Near or Remote Oceania), suggesting the origin of the first settlers of the Mariana Islands lies in the Philippines.

### Genetic relationships and burial patterns

Finally, we investigated the genetic relationships between the individuals for Nebira 2, the only site where familial relationships could be addressed (Extended Data Fig. 7 and Supplementary Table 14). The Nebira 2 cemetery contained primary burials, some of which were used as multiple or simultaneous burials[67]. People were interred in a supine extended position, mostly in western or northwestern orientation. The resulting relatedness network shows that the first- and second-degree relations at the site include both genetically male and female individuals, showing no clear signs of patri- or matrilocality. Genetically related individuals are not interred together but are buried close by, whereas unrelated individuals can be found in multiple or simultaneous burials. These results suggest that the reason for multiple burials or reusing graves was not necessarily family related or that perhaps family was not restricted to genetic relatedness as is often the case in PNG social contexts, that is, local Motu and Koita groups who inhabit the region today[68].

Analysis of runs of homozygosity (RoH) can potentially reveal higher levels of consanguinity and approximate the effective population size, which in a simplified scenario reflects the number of individuals reproducing, not the total number of individuals. The RoH show that the population at Nebira had lower levels of consanguinity and an estimated effective population size of 800–3,200 (Extended Data Fig. 8). For Eriama, a large number of short RoH is suggestive of higher background relatedness and a smaller effective population size of ~400–1,600.

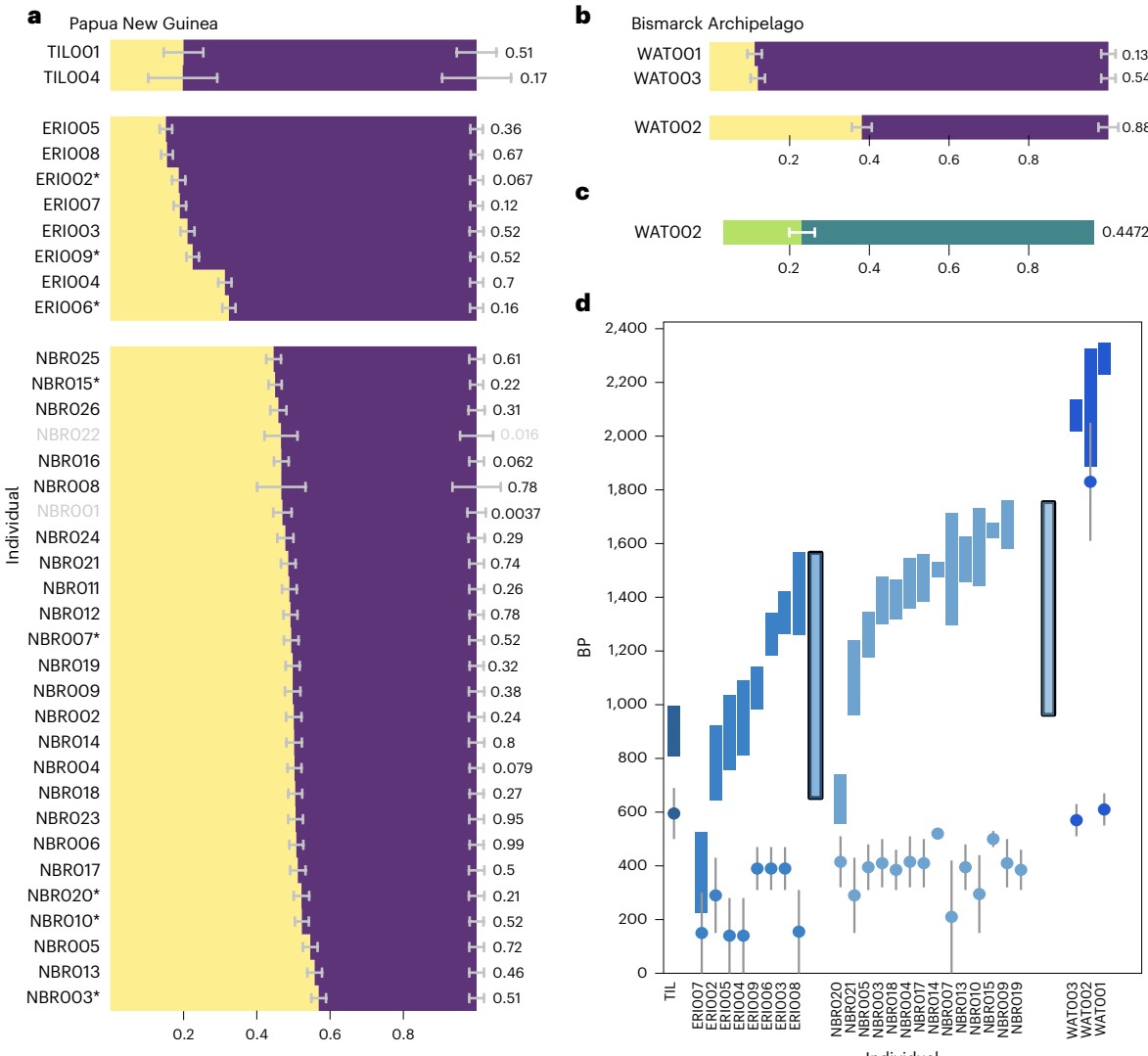

**Fig. 3 | Ancestry modelling and admixture dating.** Ancestry modelling with qpWave using Ami (yellow) and New Guinea (purple). **a,b**, Representing the ancestral components for all individuals from Papua New Guinea (**a**) and for Watom Island (**b**). **c**, Ancestry modelling for WAT002 using Lapita-related individuals from Vanuatu and Tonga (green) and an individual from Vanuatu with Papuan ancestry closely related to present-day Baining Marabu from the Bismarck Archipelago (TAN002, teal green). White lines indicate the standard error for each component, *p* values are indicated on the right-hand side of each bar, values above 0.05 are considered working models, non-working models are shown in grey font. **d**, Inferred dates of admixture (entire range indicated by filled blocks) and the midpoint of the calibrated AMS date range, based on the 95% probability ranges, for each individual (circles). Grey lines indicate the full AMS range.

The analysis of segments in the genome that are identical by descent (IBD) reveals distant genetic relations between various individuals from Eriama and Nebira (Supplementary Table 14, Extended Data Fig. 7 and Supplementary Fig. 7). The number and size of segments shared between multiple individuals from the two sites and a two-population split model suggests the two groups ceased contact around 13 generations prior, resulting in an estimated split of ~640 years ago (Supplementary Information).

## Discussion

### Time transect in Watom

The Reber–Rakival (SAC) site on Watom Island has a well-established sequence, providing evidence for occupation by people of the Lapita cultural complex from 2,800 to 2,350 BP, the middle and late Lapita phases in the Bismarck Archipelago[46,69]. The lowermost layer (Layer D) presents evidence for an earlier occupation[70]. Ceramics and obsidian are absent in this earliest layer, possibly pointing to a different population occupying the island before the arrival of the Lapita cultural complex or, potentially, co-occupation with Lapita communities during the middle and late Lapita periods on the island. At other sites on the island (SAD, SDI, SAB) both dentate-stamped Lapita pottery and sherds with opposed pinching nail impression decoration are found in secure contexts that date to before 3,000 BP (middle Lapita period).

The first analysis of ancient genomes from the Bismarck Archipelago show diverse ancestries on Watom, changing through time. WAT005/B14 and WAT006/B15 show an occupation by people with only Papuan (Baining-like)-related ancestry during the middle to late Lapita period (about 2,700–2,200 BP), at least 600 years after the archaeologically attested arrival of the Lapita cultural complex in the Bismarck Archipelago about 3,300–3200 BP (Fig. 2a and Extended Data Fig. 3a). The ⁸⁷Sr/⁸⁶Sr ratio of WAT006/Burial 15 indicated this individual was probably local to the island, as were the other individuals recovered from the site (Extended Data Fig. 1e). The archaeological and anthropological analysis of WAT006/B15 provides clues that support

the different ancestry compared to the later individuals. WAT006/B15 exhibited a rare case of cultural cranial deformation to elongate the cranium (details in the Supplementary Information and Supplementary Fig. 2)—a practice not currently known from the Lapita Cultural Complex but observed in the Aware region of southwest New Britain during the historic period[71,72]. WAT006/B15 was also the only definitely seated burial, with their head fallen into their lap. Although WAT005/B14 was disturbed with their lower half extending into the baulk, it could be determined that the individual was interred with their upper body in a supine extended position and their head oriented west to northwest. Supine extended and supine with limbs semi-flexed was, with a few exceptions, the general pattern of the later burials at the site (Petchey et al.[70]).

By 1,900 BP, we find a genetically female individual (WAT002/B10) who fits the model of mixture between Papuan-related and East-Asian-related ancestry, similar in the proportions to the individuals from the late Lapita period in Vanuatu (Extended Data Fig. 3b). Additionally, WAT002/B10 shows higher affinity to present-day non-Austronesian-speaking Baining from New Britain, similar to contemporary individuals from Vanuatu (Extended Data Fig. 3a). The connection to Vanuatu is mirrored in the incised and applied relief pottery from Watom, which shares similarities with 'Mangaasi'-style pottery from Vanuatu[73] and was found alongside dentate Lapita pottery at the Watom SDI location in a layer dating to 2,210–1,755 BP. At the Watom SAB site, no Lapita pottery but two types of locally made incised and applied relief style pottery were found in stratigraphic contexts dating to 2,455–2,310 cal BP. There may be some connection between the later arrival of Lapita individuals to Watom and incised and applied relief pottery in the late Lapita sequences on the island. Finally, the two most recent individuals from Watom (about 670–510 BP) carry a genetic make-up very similar to Austronesian-speaking Tolai people living in the region today (Fig. 2). However, they are not consistent with entirely deriving from that population (Supplementary Table 12). This is an expected observation when considering the island of Watom was highly affected by natural disasters[74,75] and colonialism that incited population mobility[76]. Evidence for population movements within the last ~500 years also comes from the oral traditions of the Tolai, which records that they came from southern New Ireland[77]. This origin is supported by the belief system and other cultural practises the Tolai share with groups in southern New Ireland but not with other groups in the Willaumez Peninsula[72].

### Proxy information regarding the settlement of the Mariana Islands

The branching pattern of the East-Asian-related ancestry of the admixed individual from Watom in a tree-like model (WAT002/B10; Extended Data Fig. 6) favours a settlement of the Mariana Islands from island Southeast Asia. The favoured route requires sailing against prevailing winds and currents and excludes a less challenging passage from the islands northeast of PNG. The lack of adequate ancient samples from the Philippines and Taiwan leaves some uncertainty to the exact geographic location from where the initial settlement of the Mariana Islands started, but regardless, expert navigation skills have to be assumed for the populations settling the islands in northwestern Remote Oceania.

### Population history of coastal PNG

The genetic affinities of the inhabitants of the southern coast of PNG show that the occurrence of pottery associated with the Lapita Cultural Complex was probably linked to the arrival of populations with East-Asian-related ancestry. All individuals from the coast of PNG harbour East Asian ancestry not observed in present-day individuals from the highlands of New Guinea but present in coastal groups in PNG today (Fig. 2a and Extended Data Fig. 1)[78]. Whereas both the Eriama and Nebira 2 sites on the southern coast are geographically close, their genetic composition suggests different population histories. On average, the individuals from Eriama show higher Papuan-related ancestry of ~80%, albeit with higher variation and younger admixture dates ranging from 1,600 to 600 BP (Fig. 3d) compared to the individuals from Nebira 2. The latter are more homogeneous, with higher genetic variation, higher proportions of East-Asian-related ancestry of > 50% and, on average, older admixture dates ranging from 1,800 to 900 BP (Fig. 3a,d,e).

The two individuals from the site of Tilu, situated on the northeastern coast of New Guinea, show a genetic profile and admixture dates similar to Eriama (Fig. 3a,d and Supplementary Table 13). First evidence for sustained settlement of the site securely dates to 650 BP, with possible earlier traces of habitation 900–800 BP[79], by people possibly speaking Austronesian languages of the Ngero–Vitiaz network[37,80]. The admixture event inferred from the individuals from Tilu falls around the time of first occupation, suggesting admixture occurred during initial settlement of the site or establishment by an already admixed population. Archaeological research places the site of Tilu in a local trading network extending to the Willaumez Peninsula on the northwestern coast of New Britain in the Bismarck Archipelago[81]. Linguistic evidence and oral traditions suggest an origin in the Vitiaz Strait in the Bismarck Sea for the Ngero–Vitiaz languages of the Bel subgroup spoken today in the area of the Tilu site, Gedaged and Bilbil[80,82,83]. However, the material exchange seems not to have extended to genetic exchange, as there is no higher affinity of the individuals of Tilu to the populations of the Bismarck Archipelago, compared to Eriama (Supplementary Table 11 and Extended Data Fig. 3a).

### Timing and genetic impact of regional dispersals

The date of ~2,100 BP for admixture between the two ancestries for all individuals from Watom with East-Asian-related ancestry suggests that admixture with local Papuan people either occurred repeatedly starting with the arrival of the Lapita Cultural Complex or only started ~1,000 years after the arrival of the Lapita cultural complex in the Bismarck Archipelago (Fig. 3d). We acknowledge the limitations of interpretations deriving from only five individuals covering a wide time transect. As previously observed[7,9], repeated admixture events may result in seemingly recent admixture dates. However, we would cautiously interpret the results as further support that the early settlers in Remote Oceania largely derived from a Lapita-associated population of East Asian ancestry that did not mix with local groups in Near Oceania. A local admixture upon or shortly after arrival in the Bismarck Archipelago would have resulted in a much earlier admixture date of ~3,000 BP. Additionally, the presence of individuals with entirely Papuan-related ancestry in the late Lapita period on Watom is consistent with the observation that the oldest individuals with exclusively Papuan-related ancestry reached Vanuatu ~2,600 BP, around 300–400 years after the initial settlement of the archipelagos[9,10,84].

The admixture dates inferred from the dated individuals on the south coast show this admixture event for people in Nebira 2 occurred at around 1,100–1,500 BP, much later than in other places such as Vanuatu (~2,600 BP) and Watom (~2,100 BP). This could be interpreted as the result of a later arrival of the descendants of people associated with the Lapita Cultural Complex along the southern coast of PNG. However, evidence for the first occurrence of Lapita settlements in the south coast of PNG is dated to starting from ~2,900–2,800 BP[27,28], 400 years after the initial Lapita formation in the Bismarck Archipelago[16,26] and contemporaneous with archaeological evidence from Vanuatu[85]. From our analysis, it is not clear whether this late admixture date is a result of long genetic isolation of the first settlers related to Lapita Cultural Complex or the result of repeated admixture with local populations carrying varying proportions of Lapita-related ancestry. Several sites demonstrating cultural contact between the Lapita Cultural groups and local groups show that interactions started shortly after the appearance of the Lapita-associated settlers[28], rendering the isolation hypothesis less likely.

The archaeological record of the south coast of PNG shows 'pulses' of settlement intensification and ceramic abundance[86]. This period, coined the 'Ceramic Hiccup'[41] but also known as 'Papuan Hiccup'[40] was possibly the result of an abandonment or relocation of sites, with a disruption in ceramic production[86,87] and styles[40,88], or the contraction of interactions to the eastern parts of the southern PNG coast[42,89], where networks were maintained, although perhaps less intensive[41]. This period is believed to have lasted between ~1,200 and 500 BP, however the exact timing remains uncertain as a result of insufficient radiocarbon dating[42]. The assumed period coincides with the medieval climate anomaly (1,250–700 BP), which was possibly delayed in Oceania, however a lack of proxy sites from PNG and Western Oceania leaves the magnitude of the effect open for the specific region[90]. Still, an increase in El Niño/Southern Oscillation events[91] may have led to seasonal droughts, suggesting the changes in settlement patterns were at least partly influenced by the availability of fresh water and the viability of cultivation[22,92]. Despite whether climate had a direct effect on the coastal communities, the interruption of long-distance trade links might have led to abandonment of many long-term settlement locales, relocation further inland[40] and perhaps been a catalyst for conflict between groups. Certainly, settlement of Nebira 4, at the base of the Nebira hill, was abandoned by the time settlement of Nebira 2, on the saddle of the double-peaked hill commenced[93–95]. A shift to defensive hilltop settlement suggests that the relationships between groups might have changed, and defensive sites on hilltops became the preferred occupational area[42].

The majority of individuals from Nebira 2 show a mixture event between 1,600 and 1,100 BP, overlapping with this period. The individual NBR020/ACJ-34 shows an admixture date of only ~650 years, much younger than the majority of the group. Additionally, this individual displayed a non-local $^{87}Sr/^{86}Sr$ ratio[54] and a different childhood diet, suggesting they may have originated from a population with a different ancestral history. The admixture date of NBR020/ACJ-34 coincides with a period when coastal and inland settlements had become archaeologically visible, and ceramic wares show stylistic innovations with regional variation indicating the establishment or reinforcement of social boundaries[96–99]. Trade supposedly resumed, perhaps with reconfigured networks, in ~800–500 BP and has been associated with different Austronesian-speaking groups arriving on the coasts[78].

Overlap of maritime trade networks such as the Kula and *hiri* trade networks by at least 500 BP[42,86,100,101], suggests a (re-)emergence of these networks in the period following the 'Papuan Hiccup'[42].

Admixture dates inferred from the individuals from Eriama (Fig. 3d), together with the higher Papuan ancestry (Fig. 3a and Extended Data Fig. 3) indicate a continuation of genetic exchange beyond that identified for Nebira 2. The strong shift from 40% to 15% Asian-related ancestry suggests they were part of an interaction sphere with people carrying higher Papuan-related ancestry in the model represented by Papuan Highlanders. However, we lack the resolution to identify which populations contributed and where their geographical source was. During the 'Papuan Hiccup', communities might have retreated into the interior areas, where interactions with highland populations might have intensified[40]. One individual from Eriama (ERI007/ACV-7) shows more recent admixture inferred to around 600–400 years ago, also implying that the population in Eriama continued genetic exchange with other populations.

Despite the many differences between the two sites, analyses of IBD blocks reveal distant connections between Nebira 2 and Eriama, suggesting that they constituted one reproductive unit ~13 generations before the analysed population, estimating the split to ~640 BP. The different genetic compositions of the two nearby sites show that the southern coast was a genetic, and possibly also a cultural and linguistic mosaic of people, matching the situation in Motu and Koita speaker communities today. The Motu language is a Central Papuan Tip language of the Western Oceanic branch of the Oceanic group of Austronesian languages[102,103] and is spoken mostly by people living in the coastal region of the Central Province. The Papuan language dominant in the region, Koita[104], is spoken in settlements more inland[38,39,105,106]. The two major cultural groups occupying the coastal and interior areas in the vicinity of Nebira 2 and Eriama have historically close social connections that regularly included intermarriage, with both groups involved in the maritime *hiri* trade expeditions[107]. Differences in burial practices, specifically primary burial at Nebira 2 and secondary cave burial at Eriama, also point to distinct cultural practices regarding the treatment of the dead.

The oral traditions of present-day groups remembering their ancestors residing at Nebira 2 and Eriama include tales of a relocation from the mountains[67,105]. The reason for the separation of the two communities, despite no evident geographical barriers, remains enigmatic and might point to the introduction of a cultural barrier connected to different interactions spheres of the two groups after relocation.

## Methods

### Sample processing

All samples were processed in dedicated ancient DNA laboratories at the Max Planck Institute for the Science of Human History (now MPI for Geoanthropology) in Jena, Germany.

Bone powder from the petrous part of the temporal bone was obtained through cutting along the margo superior partis petrosae (crista pyramidis) and drilling 50–150 mg bone powder from the densest part around the cochlea[108]. Teeth were sampled by cutting along the junction of the root and the crown and drilling ~50 mg from the pulp chamber. In total, 46 samples were destructively sampled of which 41 could be included in the analysis.

DNA extraction was carried out following established protocols[109]. Negative and positive controls were included. To release DNA from 50–100 mg of bone powder, a solution of 900 µl ethylenediaminetetraacetic acid (EDTA), 75 µl $H_2O$ and 25 µl Proteinase K was added. In a rotator, samples were digested for at least 16 h at 37 °C, followed by an additional hour at 56 °C (ref. 110). The suspension was then centrifuged and transferred into a binding buffer as previously described[109]. Samples were purified over silica columns for high volumes (High Pure Viral Nucleic Acid Large Volume Kit; Roche) with two washing steps using the manufacturer's wash buffer. DNA was eluted in TET (10 mM Tris, 1 mM EDTA and 0.05% Tween) in two steps for a final volume of 100 µl.

Double-stranded DNA libraries were built from 25 µl of DNA extract in the presence of uracil DNA glycosylase (UDGhalf libraries), following a double-stranded (ds)'UDG-half' library preparation[111]. Negative and positive controls were carried alongside each experiment. Libraries were quantified using the IS7 and IS8 primers[112] in a quantification assay using a DyNAmo SYBP Green qPCR Kit (Thermo Fisher Scientific) on the LightCycler 480 (Roche). Each aDNA library was double indexed[113] in 1–4 parallel 100 µl reactions using PfuTurbo DNA Polymerase (Agilent). The indexed products for each library were pooled, purified over MinElute columns (Qiagen), eluted in 50 µl TET and again quantified using the IS5 and IS6 primers[112] with the quantification method described above. Four µl of the purified product were amplified in multiple 100 µl reactions using Herculase II Fusion DNA Polymerase (Agilent) following the manufacturer's specifications with 0.3 µM of the IS5/IS6 primers. After another MinElute purification, the product was quantified using the Agilent 2100 Bioanalyzer DNA 1000 chip. An equimolar pool of all libraries was then prepared for shotgun sequencing on Illumina platforms in 75 base pair single-end-run cycles using the manufacturer's protocol. For nine individuals with low DNA content in ds libraries, we produced a second, single-stranded (ss) library[114] with an automated protocol as detailed in ref. 115.

Libraries were further amplified with IS5/IS6 primers to reach a concentration of 200–400 ng µl$^{-1}$ as measured on a NanoDrop spectrophotometer (Thermo Fisher Scientific). Mitochondrial DNA capture[116] was performed on screened libraries which, after shotgun sequencing,

showed the presence of aDNA, highlighted by the typical CtoT and GtoA substitution pattern towards 5′ and 3′ molecule ends, respectively. Furthermore, samples with a percentage of human DNA in shotgun data 0.1% or greater were enriched for a set of 1,237,207 targeted SNPs across the human genome[117]. The enriched DNA product was sequenced on an Illumina HiSeq 4000 instrument with 75 single-end-run cycles using the manufacturer's protocol. The output was de-multiplexed using bcl2fastq version 2.17.1.14 (Illumina conversion Software) and dnaclust version 3.0.0[118].

Pre-processing of the sequenced reads was performed using EAGER version 1.92.55[119]. The resulting reads were clipped using *Clip&Merge*[119] and *AdapterRemoval* version 2[120]. Clipped sequences were mapped against the human reference genome hg19 using the Burrows–Wheeler Aligner (BWA) version 0.7.12[121] disabling seeding (−l 16500, −n 0.01). Duplicates were removed with DeDup version 0.12.2[119]. A mapping quality filter of 30 was applied using SAMtools version 1.3[122]. In ds libraries, reads were trimmed for two base pairs. Different sequencing runs and libraries from the same individuals were merged, duplicates removed and sorted again using SAMtools[122]. Trimmed and untrimmed reads were genotyped separately using pileupCaller version 8.6.5 (https://github.com/stschiff/sequenceTools). We combined the genotypes keeping all transversions from the untrimmed genotypes and transitions only from the trimmed genotypes to eliminate problematic, damage-related transitions on the ends. ss-libraries were genotyped based on the untrimmed reads using the –singleStrandMode. The generated pseudo-haploid calls from both ss and ds libraries were merged using a custom python script, which keeps all identical positions across the two genotypes and the sites covered only in one of the two. For sites covered in both libraries, but with different base calls, the state of the genotype was randomly picked from one of the libraries. The final genotypes of all ancient individuals were merged to a pulldown of the 1,240 K SNPs from the Simons Genome Diversity Project[123], a set of individuals from Asia and the Pacific, genotyped on the Human Origins array[124], on the Illumina Infinium Multi-Ethnic Global Array[100] and ancient Asian and Oceanian individuals[8,10,66,124–127].

The typical features of ancient DNA were inspected with *DamageProfiler* version 0.3.1 (http://bintray.com/apeltzer/EAGER/DamageProfiler)[119] (Supplementary Table 8). Sex determination was performed by comparing the coverage on the targeted X-chromosome SNPs (~50 K positions within the 1,240 K capture) normalized by the coverage on the targeted autosomal SNPs to the coverage on the Y-chromosome SNPs (~30 K), again normalized by the coverage on the autosomal SNPs[128] (Supplementary Table 1). For male individuals, ANGSD version 0.919[129] was run to provide an estimate of nuclear contamination in males. All male samples with at least 100 X-chromosome SNPs covered twice exhibited X-chromosome contamination levels below 7%, hence all reads were retained for further analyses (Supplementary Tables 1 and 4). For both genetically male and female individuals, mtDNA-captured data were used to jointly reconstruct the mtDNA consensus sequence and estimate contamination levels with Schmutzi[130] (Supplementary Table 1) and used as reliable predictors for nuclear contamination[131,132]. The software ADMIXTURE version 1.3.0 (69) was used in unsupervised mode to allow for free genetic clustering with a worldwide set of individuals[123,133–135] (Extended Data Fig. 1).

## AMS and isotope analysis

The 14 new radiocarbon dates for this study and the stable isotope analysis was produced at the Curt–Engelhorn–Zentrum Archäometrie gGmbH in Mannheim, Germany. Details on the protocols can be found in the Supplementary Information. Marine reservoir correction was applied to correct for the offset in 14 C between the atmosphere and the oceans ($\Delta R$)[45–49]. $\Delta R$ was applied to the most recent marine calibration curve[136] after determining the percentage of marine food in the diet from the bone stable isotope values[46,49] (Supplementary Tables 2–4). For individuals from Eriama and Nebira, fully terrestrial calibrations

were produced. For Burials 1 and 12 (WAT001/003) from Watom, a marine deltaR of 172 ± 72 for shell/mixed diet animals was applied using the marine20 calibration curve and for the individual from Tilu, a $\Delta R$ of −111 ±16 and 28 ± 10 marineC[137]. For burials 15, 10 and 14 (WAT002, 005 and 006) from Watom, a marine carbon contribution of 28 ± 10% marineC was used in the calibration.

Approximately 0.5–0.8 g of cortical bone was sampled for the stable isotope analysis[50,51]. For the dentine stable isotope analyses, the distal third of the root of the permanent first molar (~0.2–0.3 g, measured from the cemento–enamel junction, formation time between the ages of 5–9 years)[138,139] was sampled with a Dremel drill fitted with a diamond edge saw. The dentine was sampled horizontally rather than along growth increment lines for the purposes of attaining enough sample to analyse[140]. Secondary dentine was removed from inside pulp cavity of the tooth root with a Dremel drill fitted with a diamond burr. The samples of dentine were then sonicated for five minutes and fully dried.

Bone and tooth samples were cleaned with alum oxide air abrasive equipment (Bego Easyblast). A modified Longin method[141] was used to extract collagen from the bone and dentine samples at the University of Otago, Dunedin, New Zealand[142,143]. All bone and dentine samples were soaked in 0.5 M HCl at 4 °C (changed every other day) until completely demineralized. The demineralized samples were then rinsed in MilliQ $H_2O$ until they reached a neutral pH. The samples were gelatinized at 70 °C in a pH 3 solution for 48 h, followed by filtering with 5–8-µm Ezee mesh filters (Elkay Laboratory Products) to remove any reflux-insoluble residues and then ultrafiltered with Millipore Amicon Ultra-4 centrifugal filters (30,000 nominal molecular weight limit (NMWL)) to retain molecules larger than 30 kDa (ref. 144). The purified 'collagen' was frozen and then lyophilized for 48 h and subsequently weighed into tin capsules before analysis by elemental analyser–isotope ratio mass spectrometry (EA–IRMS) at either IsoTrace (Dunedin, New Zealand) or Iso-Analytical (Cheshire, UK)[50,51]. Analytical error, calculated from replicate measurements of samples, was ± 0.1‰ for $\delta^{13}C$ and ± 0.2‰ for $\delta^{15}N$ (1 standard deviation (SD)).

The stable isotope analysis of the petrous bones analysed for AMS dating was conducted at Curt–Engelhorn–Zentrum Archäometrie gGmbH, Mannheim by the same method described above, except for the addition of a step to remove humic acids with NaOH after demineralization. Analyses were conducted in triplicates and included (1) combustion of the sample and determination of C and N per cent using a vario PYRO cube CNSOH elemental analyser (Elementar); (2) determination of isotope ratios using a precision isotope ratio mass spectrometer (Isoprime); (3) data are corrected to USGS 40 and USGS 41a using the internal software (two-point-normalization).

For the enamel carbonate analyses, the enamel surface was cleaned of surface contaminants by abrasion with a Dremel rotary tool with a diamond cutting blade, and any adhering dentine was removed. Two enamel chips weighing between 10–50 mg a piece were sampled for carbon and oxygen stable isotope analysis[54] and strontium isotope analyses (below)[54]. For the carbon and oxygen isotope analyses (note that oxygen results not reported here), the enamel chip was ground in a clean mortar and pestle and transferred into an acid-cleaned glass vial. Following pretreatment protocols[145] the samples were first soaked in 2% NaOCl (sodium hypochlorite) for 24 h, rinsed with MilliQ $H_2O$ and then soaked in 0.1 M acetic acid for another 24 h to remove organic and secondary diagenetic carbonates. Carbon stable isotope analyses from the carbonate were undertaken at the Isotrace Research Laboratory, Dunedin, New Zealand, using a Thermo Delta Plus Advantage linked to a Gasbench II via a GC PAL autosampler. Delta values were normalized and reported against the international standards Vienna Pee Dee Belemnite. Analytical precision was determined through repeated analysis of laboratory standards IA-R022 (calcium carbonate), NBS-18 (calcite) and NBS-19 (calcite) and duplicate samples. Analytical error, calculated from replicate measurements of samples, was ±0.1 for $\delta^{13}C$ (1 SD, $n = 10$).

 

For strontium isotope analysis, ion exchange techniques were used to isolate and purify the Sr fraction from the digested sample matrix at the Centre for Trace Element Analysis, Department of Geology, University of Otago. Enamel chips weighing 10–20 mg were sonicated in MilliQ H$_2$O, dried and transferred into clean perfluoroalkoxy alkanes vials (Savillex) and weighed in the clean lab before digestion in 2 ml of 3 M HNO$_3$ solution at 110 °C overnight. Once fully digested, samples were evaporated for four hours at 110 °C. Strontium was manually separated utilizing a micro-chromatographic exchange column, Eichrom Sr-SPEC resin, and the established method of column chemistry[146]. Only a single elution was necessary for the human samples which were then evaporated and dissolved in a 2% HNO$_3$ solution for mass spectrometric analysis.

The $^{87}Sr/^{86}Sr$ values were measured using a Nu Plasma-HR MC-ICP-MS instrument (Nu Instruments Ltd., UK). The $^{87}Sr/^{86}Sr$ data were normalized using repeated measurement of the NIST-SRM 987 standard ($n = 25$, average $^{87}Sr/^{86}Sr = 0.710286 \pm 0.000013$ (1 SD) in very good agreement with the accepted value of $0.71034 \pm 0.00013$ (1 SD)[147]. An in-house sample of a giant clam (*Tridacna*) (ANU) carbonate control ($n = 5$, average $^{87}Sr/^{86}Sr = 0.70920 \pm 0.0000061$ SD) which is consistent with expected seawater $^{87}Sr/^{86}Sr$ value of 0.7092[148]. Total procedural blanks for the chemical separation process were 60 ng which is negligible relative to the amount of Sr in the tooth enamel samples. Duplicate analyses were also performed on three samples which ranged between ±0.000013 and 0.000027 (2 SE). All uncertainties are reported at the 2-sigma level, unless stated otherwise.

### Population genomic analysis

Principal component analyses were performed using *smartpca* version 13050[149] with a set of populations from East Asia and the Pacific[78,100,123,133–135] (Fig. 2 and Supplementary Table 10). Ancient individuals[8,10,66,124–127] were projected onto the calculated components using the options 'lsqproject: YES', 'shrinkmode: YES' and 'numoutlieriter: 0'. Individuals with less than 20,000 SNPs covered were not projected with the exception of WAT006.

To identify the differences on an individual basis and we used *qp3Pop* version 5.0 (70) and computed $f_3$-outgroup statistics comparing all individuals to each other with Mbuti.DG serving as an outgroup. We used *qpDstat* version 5.0[133] to run $f_4$-statistics of the form $f_4$(*Mbuti, Ami. DG Individual 1 site X; Individual 2 site X*) (Supplementary Table 10). The values close to zero for all individuals excavated from Nebira suggested a grouping of the individuals by site was reasonable for certain analyses. For individuals excavated from Eriama, the two individuals ERI004 and ERI006 produced absolute Z-scores greater than 3 with all other individuals from the site and hence were kept as a separate group. Both individuals excavated from the site Tilu and the two younger individuals excavated on Watom (WAT001 and WAT003) were also grouped on this basis, whereas WAT002, WAT005 and WAT006 were kept separate, also accounting for the long-time intervals between them (Table 1). To test which present-day populations represented best the Papuan and East-Asian-related ancestries in the individuals, we computed $f_4$ statistics of the form $f_4$(*Mbuti.DG, Test, X, New_Guinea*) and $f_4$(*Mbuti.DG, Test, X, Ami*), respectively, testing in X all other populations from the region (Supplementary Table 11). To understand whether the Asian ancestry component was more similar to the Early Remote Oceanians (ERO) from Vanuatu and Tonga[124] compared to ancient Austronesians from Taiwan (Suogang)[125], we calculated $f_4$(*Mbuti.DG, Test; Suogang, ERO*), expecting positive test scores for a higher affinity to ERO (Supplementary Table 11 and Extended Data Fig. 3b). To understand the differential affinities with respect to Near Oceanian populations, disregarding the differences in Asian ancestry, we produced a scatterplot (Extended Data Fig. 3a) based on the $f_4$(*Mbuti. DG, Ami, Test, Baining_Marabu/New_Guinea*).

We used qpWave version 410[150] to test whether individuals were consistent with deriving from the same group as other individuals from the same site, relative to a set of reference groups (Mbuti, Onge, New_Guinea, Baining_Marabu, Ami, Han, English, Chukchi, Nasioi, Denisova_published.DG). To test whether some of the individuals could be modelled as consisting of a single ancestry component, we modelled the respective individual and Ami to test for exclusively Asian ancestry and New_Guinea Highlanders to test for exclusively Near Oceanian ancestry (Supplementary Table 12) using the same references detailed above, excluding the respective populations used in the test[151]. After identifying the individuals not consistent with deriving from one respective ancestry, we used qpAdm version 5.0[133] to model all groups and the individuals in each group covered by more than 50,000 SNPs as a two-way admixture between New_Guinea Highlanders and Amis (Supplementary Table 12) and the grouped individuals as a mixture of Early Remote Oceanians and TAN002, a previously published individual with exclusively Baining-like, ancestry[9]. As reference groups, we used Mbuti, Onge, Han, Chukchi, English and Denisova_published.DG. To test for sex-biased admixture, we calculated the excess ancestry on the X chromosome restricting the above analysis on grouped individuals restricted to the X chromosome (Supplementary Table 12). We subtracted the value obtained from all chromosomes from that obtained from the X chromosome alone (Supplementary Table 12).

To evaluate the genetic affinity relative to other selected populations in a tree-like representation, we used qpGraph version 6450, with the parameters outpop: NULL, useallsnps: NO, blgsize: 0.05, forcezmode: YES, lsqmode: YES, diag: .0001, bigiter: 6, hires: YES, lambdascale: 1. We used this tool strictly for testing two hypotheses: (1) the genetic composition of the individuals from Guam and Saipan[8,66], in this model grouped into one 'Mariana' group, is more similar to that of Austronesian populations in Taiwan or the Philippines, suggesting the Marianas were settled directly from Island South East Asia; (2) the ancestry is more similar to the Asian component of WAT002, suggesting that the expansion first arrived in the Bismarck Archipelago, and from there, facilitated by the prevailing winds and currents, travelled to the Mariana islands. For completeness of possible scenarios, we also tested (3) if the ERO component in Remote Oceania is ancestral to Watom and Guam/Saipan. To test these three models, we first constructed a base tree including Mbuti.DG, Ami. DG, Igorot.DG, Papuan.DG[123], then adding according to their chronological age ERO[9,124], followed by the ancient genomes from Guam and Saipan[8,66]. The best-fitting base tree models Ami.DG and Igorot.DG as sister groups shows no admixture events and is in concordance with the genetic data, reflected in a Z-Score of $|Z| = 2.8$ (Extended Data Fig. 6a). Trees where the Marianas were modelled to split with Amis. DG (suggesting settlement from Taiwan) or with Igorot.DG (from the Philippines) were rejected ($|Z| = 10.88$, $|Z| = 11.303$, respectively; Extended Data Fig. 6d,e). We fitted B10/WAT002 as a mixture of Papuan.DG and an Asian-related component deriving from different branches reflecting the patterns consistent with the respective hypothesis: assuming (1), B10/WAT002s Asian-related component was fixed to split as a sister group of the ERO genomes. In the alternative hypothesis (2), the Asian-related component of WAT002 was fit as branching after Ami.DG and Igorot.DG, resulting in sister groups of Guam/Saipan and ERO. The first (1) tree (Extended Data Fig. 6b) provides a better fit for the data, indicated by the worst Z-score of $|Z| = 2.883$. The fit of the second (2) tree (Extended Data Fig. 6c) is much worse ($|Z| = 4.196$) and shows a zero-drift branch from the common ancestor of ERO and the Marianas, and the ancestry present in WAT002, suggesting a poor fit. The model (3), suggesting the East-Asian-related ancestry arrived first in Remote Oceania before dispersing back to the Bismarck Islands and the Marianas, also shows a poor fit ($|Z| = 4.196$) (Extended Data Fig. 6f).

To assess the maternal lineage, the mtDNA-enriched sequences were processed in nf-core/eager version 2.4.0[152] (https://nf-co.re/eager) using Nextflow version 21.04.3[153]. FastQC version 0.11.9 (https://www.bioinformatics.babraham.ac.uk/projects/fastqc/) and fastP version

0.20.1. Fastp[154] was used for sequencing quality control, adaptors were removed with AdapterRemoval version 2.3.2[120]. The remaining reads were aligned to the mitochondrial reference genome with circular mapper version 1.0[119] and the resulting bam files filtered with SAMtools version 1.12[122] for a minimum mapping quality and minimum read length of 30. Qualimap version 2.2.2-dev[155] and bedtools version 2.30.0[156] were used to generate mapping statistics and duplicates were removed using Picard MarkDuplicates version 2.26.0 (http://broadinstitute.github.io/picard/). Ancient DNA damage was assessed with DamageProfiler version 0.4.9[157], endogenous DNA estimated using endorS.py version 0.4 (https://github.com/aidaanva/endorS.py) and the final report generated with MultiQC version 1.11[158]. From the resulting bam files, we built consensus sequences using the export function in Geneious version 2019.2.3[159], setting 'If no coverage call' and to 'Call '{}' if coverage < '{}''N/X with a coverage threshold of 5, aligned to 'highest quality' and >50% Sanger heterozygotes. Only NBR008 did not contain enough reads for sequence calling. For all other individuals, mitochondrial haplogroups were then determined with HaploGrep version 2.4.0[160] Most individuals were assigned to 'Austronesian'-associated haplogroups with 23 B4a1a*1, four B4a1a1a and one B4a1a1 haplogroup assignment. For NBR006 we identified haplogroup F1a3a, associated with present-day populations from the Philippines. For NBR017 the haplogroup P1, frequently found in Papua New Guinea and for NBR004 and NBR013 the Papuan-associated haplogroup Q1. Haplogroups could not confidently be assigned for NBR022, NBR008 and WAT006. Y-chromosomal haplogroups were identified by calling the SNPs covered on the Y chromosome of all male individuals by using the pileup from the Rsamtools[161] package and assigning haplogroups by analysing the overlap with the ISOGG SNP index v.14.07 as detailed in (Extended Data Fig. 5)[162].

To evaluate parental relatedness, runs of homzygosity (ROH) (Extended Data Fig. 8) were detected using hapROH v.0.3a4[163] in Jupyter notebooks v.6.4.4. Using the provided down-sampled 1,000 Genomes data as a reference panel, the habs_ind command evaluated ROH on 22 chromosomes (chs=range(1,23)) with data-specific parameters (e_model = 'haploid', p_model = 'Eigenstrat', n_ref=2504, random_allele=True, readcounts=False, delete=False, logfile=True, combine=True) (Extended Data Fig. 8). The ROH were also used to estimate effective population size ($N_e$) with a maximum likelihood approach (MLE_ROH_Ne) after removing individuals with ROH segments >20 cM (indicative of very close parental relatedness).

The genetic relatedness of all individuals within a site was calculated using READ[164] (Supplementary Table 14). Using this signal, we identified first- and second-degree relationships, illustrated in Extended Data Fig. 7. To corroborate these results and understand more distant genetic relatedness, genetic segments identical by descent (IBD blocks) were analysed. To prepare the data, ATLAS v.0.9 was used to call genotype likelihoods (method = MLE) for all positions from the 1,000 Genomes Phase 3 release after recalibrating the base quality scores (regions = '88_mammals.epo_low_coverage.10M_GRCh37.masked.bed') according to post-mortem damage (length = 50). The genotype likelihoods were then used for imputation with GLIMPSE v.1.0.0, followed by sub-setting to the 1,240 K SNPs. Imputation quality was assessed by counting the number of SNPs with genotype probabilities above 0.99 and excluding individuals with less than 600,000 high-quality SNPs (ERI003, ERI004, NBR001, NBR002, NBR005, NBR007, NBR008, NBR009, NBRO14, NBR017, NBR020, NBR022, NBR024, NBR026, WAT002, WAT005, WAT006). Finally, ancIBD v.0.2a2 (https://pypi.org/project/ancIBD/)[165] was used to calculate IBD blocks per individual from the 1,240 K-extracted data using the recommended default settings. We filtered to IBD > 20 cm to detect relationships up to the sixth degree that typically share multiple such long IBD segments. The IBD segments were further integrated to a model calculating the split times between the two sites. Details on the calculations can be found in the Supplementary Information.

## Ethics statement

Permissions for the samples included in this study were provided by the National Museum and Art Gallery of Papua New Guinea (NMAG) to H.B. for Nebira, Eriama and Watom and G.S. and D.G. for Tilu and Nunguri, including the ancient DNA processing. The scope of the study and the results were discussed with members of the NMAG before submission and shared with the Papua New Guinea Institute of Medical Research for consideration through their Institutional Review Board. Results of the study are communicated to the general public in Madang through personal communication of G.S. and materials made available to the NMAG, Madang University and various local high schools. The skeletal elements will be returned to the NMAG (Nunguri and Tilu) and to the Department of Anatomy, University of Otago, Aotearoa (Nebira, Eriama and Watom).

## Reporting summary

Further information on research design is available in the Nature Portfolio Reporting Summary linked to this article.

## Data availability

The raw data of the captured libraries are available at the European Nucleotide Archive (ENA) at https://www.ebi.ac.uk/ena/browser/view/PRJEB68153 under the accession number PRJEB68153. The genotypes of the newly published individuals can be sourced through the Poseidon framework community archive via Github at https://github.com/poseidon-framework/community-archive under 2024_NaegeleNatureEcologyEvolution. We ask users of this data to consider the recommendations on the use detailed in the Supplementary Information, 'Ethical considerations for the analysis of ancient genomes'. The skeletal elements sampled will be returned to the National Museum and Art Gallery of Papua New Guinea in Port Moresby (Nunguri and Tilu) and to the Department of Anatomy, University of Otago, Aotearoa (Nebira, Eriama and Watom) by spring of 2025 to be curated with their respective skeletal assemblages.

## Code availability

The code used in the site split analysis is deposited via GitHub at https://github.com/hyl317/two_island.

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

## Acknowledgements

We thank the National Museum and Art Gallery of Papua New Guinea for the permission to analyse the individuals excavated from Watom, Nebira, Eriama Nunguri and Tilu. We specifically thank A. Kuaso for his consultations and input on the final results. We acknowledge the pioneering work of the late S. Bulmer. She led extensive surveys and excavations on the south coast and highlands of Papua New Guinea. We thank the Tilu community at Malmal village and the Watom communities for their assistance in the research. Many thanks to F. Aron and L. Papac for support in the wet lab. We thank S. Halcrow for assistance with the repatriation of the studied elements. This study was funded through the 'WAVES' starting grant, granted by the European Research Council (ERC) (#ERC758967) awarded to A.P., a University of Otago Research Grant, a University of Otago Doctoral Scholarship and the Max Planck Society.

## Author contributions

R.K., H.B., D.G., G.S., M.T., P.P. and D.A. contributed archaeological material and contextualized the genetic findings. R.K. analysed stable isotope data with critical input from B.S., C.S., M.R. and D.B. F.P. performed the corrections for the marine reservoir effect. M.T. analysed microparticle data. K.N., E.B. and R.R. performed ancient DNA laboratory work. K.N., Y.H., A.B.R. and S.C. performed population genetic analysis. K.N. and R.K. interpreted the data with critical input from C.P., J.K., H.R., A.B.R., archaeological contextualization by D.G., G.S., M.T., R.K., B.S. and H.B. and linguistic contextualization by M.W. K.N. wrote the paper with critical input from C.P., R.K., J.K., A.P., G.S., D.G. and the other authors. K.N., Y.H., S.C. and R.K. produced the figures. A.P., G.S., H.B. and R.K. organized sample collection. A.P., J.K. and C.P. conceived of and coordinated the study.

## Funding

## Competing interests

The authors declare no competing interests.

## Additional information

**Extended data** is available for this paper at https://doi.org/10.1038/s41559-025-02710-x.

**Correspondence and requests for materials** should be addressed to Kathrin Nägele, Rebecca Kinaston, Cosimo Posth or Johannes Krause.

Kathrin Nägele ®[1,22] ✉, Rebecca Kinaston ®[2,22] ✉, Dylan Gaffney ®[3,4], Mary Walworth ®[5,6], Adam B. Rohrlach ®[1,7], Selina Carlhoff ®[1], Yilei Huang[1,8], Harald Ringbauer ®[1,9], Emilie Bertolini[1], Monica Tromp ®[4,10], Rita Radzeviciute[1], Fiona Petchey ®[11,12], Dimitri Anson[4], Peter Petchey[13], Claudine Stirling[14,15], Malcolm Reid ®[15], David Barr[15], Ben Shaw ®[16], Glenn Summerhayes[4,17], Hallie Buckley ®[18], Cosimo Posth ®[1,19,20] ✉, Adam Powell ®[21] & Johannes Krause ®[1] ✉

[1]Department of Archaeogenetics, Max Planck Institute for Evolutionary Anthropology, Leipzig, Germany. [2]BioArch South, Waitati, New Zealand. [3]School of Archaeology, University of Oxford, Oxford, UK. [4]Archaeology Programme, University of Otago, Dunedin, New Zealand. [5]Department of Linguistic and Cultural Evolution, Max-Planck-Institute for Evolutionary Anthropology, Leipzig, Germany. [6]CRLAO (UMR8563), Centre National de la Recherche Scientifique, Paris, France. [7]School of Computer and Mathematical Sciences, University of Adelaide, Adelaide, South Australia, Australia. [8]Bioinformatics Group, Institute of Computer Science, Universität Leipzig, Leipzig, Germany. [9]Department of Human Evolutionary Biology, Harvard University, Cambridge, MA, USA. [10]Southern Pacific Archaeological Research (SPAR), University of Otago, Dunedin, New Zealand. [11]Radiocarbon Dating Laboratory, Te Aka Mātuatua – School of Science, University of Waikato, Hamilton, New Zealand. [12]ARC Centre of Excellence for Australian Biodiversity and Heritage, College of Arts, Society and Education, James Cook University, Cairns, Queensland, Australia. [13]Southern Archaeology Ltd, Dunedin, New Zealand. [14]Department of Geology, University of Otago, Dunedin, New Zealand. [15]Centre for Trace Element Analysis, Department of Geology, University of Otago, Dunedin, New Zealand. [16]School of Culture History and Language, Australian National University, Canberra, Australian Capital Territory, Australia. [17]School of Social Science, University of Queensland, Saint Lucia, Queensland, Australia. [18]Department of Anatomy, School of Biomedical Sciences, University of Otago, Dunedin, New Zealand. [19]Archaeo- and Palaeogenetics, Institute for Archaeological Sciences, Department of Geosciences, University of Tübingen, Tübingen, Germany. [20]Senckenberg Centre for Human Evolution and Palaeoenvironment, University of Tübingen, Tübingen, Germany. [21]Department of Human Behavior, Ecology and Culture, Max Planck Institute for Evolutionary Anthropology, Leipzig, Germany. [22]These authors contributed equally: Kathrin Nägele, Rebecca Kinaston. ✉e-mail: kathrin_naegele@eva.mpg.de; contact@bioarchsouth.com; cosimo.posth@uni-tuebingen.de; krause@eva.mpg.de

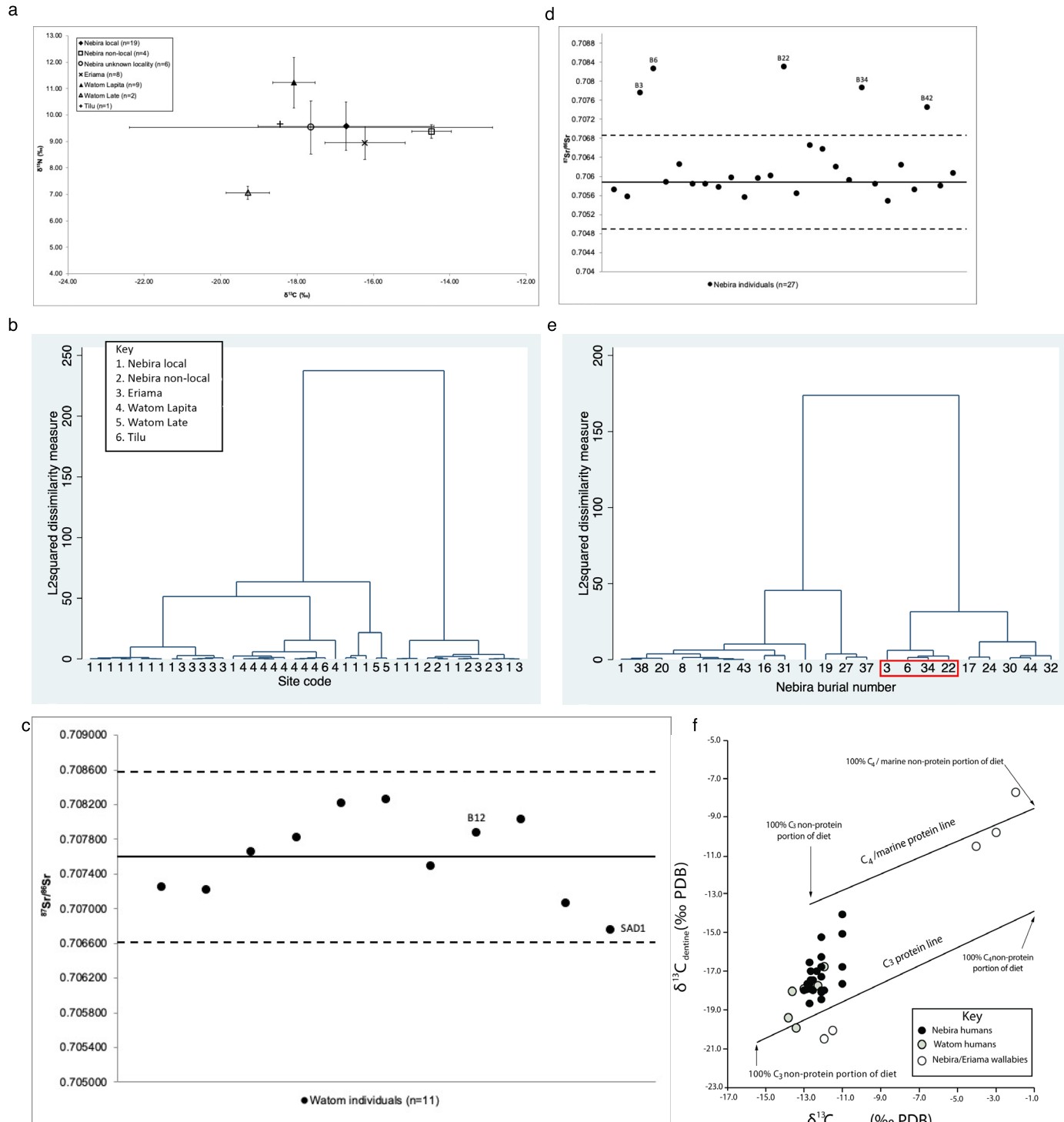

**Extended Data Fig. 1 | Isotope analysis. (a)** Comparison of mean (± 1 SD) human $\delta^{13}C_{collagen}$ and $\delta^{15}N_{collagen}$ values for each site; **(b)** Ward's hierarchical cluster analysis of the $\delta^{13}C_{collagen}$ $\delta^{15}N_{collagen}$ values of the humans from all sites; **(c)** Strontium ($^{87}Sr/^{86}Sr$) results for the humans from Watom, compared to the human mean $^{87}Sr/^{86}Sr$ value (±2 SD) from the site (the non-Lapita burials [SAD1 and B12] are identified on the graph); **(d)** Strontium ($^{87}Sr/^{86}Sr$) results for the humans from Nebira compared to the mean wallaby $^{87}Sr/^{86}Sr$ value (±2 SD)

(the non-local burials are identified on the graph). **(e)** Ward's hierarchical cluster analysis of the $\delta^{13}C_{collagen}$, $\delta^{15}N_{collagen}$, $\delta^{13}C_{dentine}$, $\delta^{15}N_{dentine}$, $\delta^{13}C_{enamel}$, and $^{87}Sr/^{86}Sr$ values of the humans from Nebira (the red rectangle delineates the non-local individuals as identified from the strontium isotope analysis, excluding Burial 42, who did not provide $\delta^{13}C_{collagen}$, $\delta^{15}N_{collagen}$ data). **(f)** $\delta^{13}C_{dentine}$ and $\delta^{13}C_{enamel}$ results for the Nebira and Watom humans and Nebira and Eriama wallabies are presented in reference to experimental dietary regression lines presented in [166].

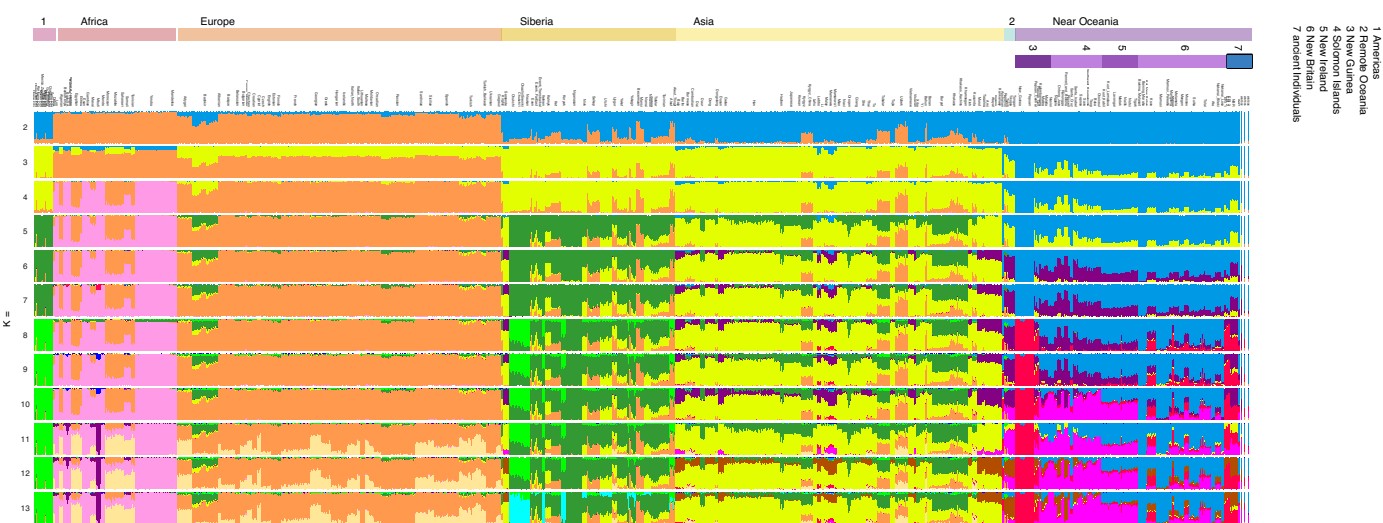

**Extended Data Fig. 2 | Genetic clustering analysis using ADMIXTURE.** Clusters K = 2-13 are computed based on the available individuals genotyped of the Human Origins array[124,133] and individuals from the Americas from the SGDP dataset[123,135].

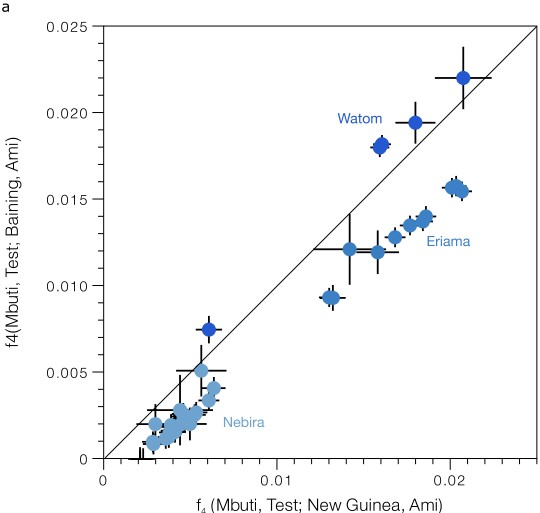

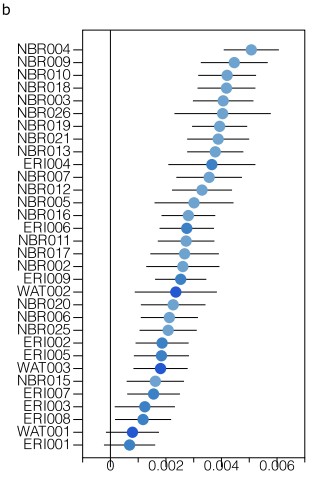

**Extended Data Fig. 3 | $f_4$-statistics. (a)** $f_4$-scatterplot investigating the affinities to New Guinean highlanders (x-axis) and Baining from New Britain in the Bismarck Archipelago (y-axis). (**b**) $f_4$-statistic showing affinity of the Asian component to ancient Taiwanese (Suogang, negative test scores) or Early Remote Oceanians (ERO) from Vanuatu and Tonga (positive test scores). Tests with overlapping SNPs < 1000 are not shown. Points indicate $f_4$-value; black lines indicate one standard error determined by block jackknive.

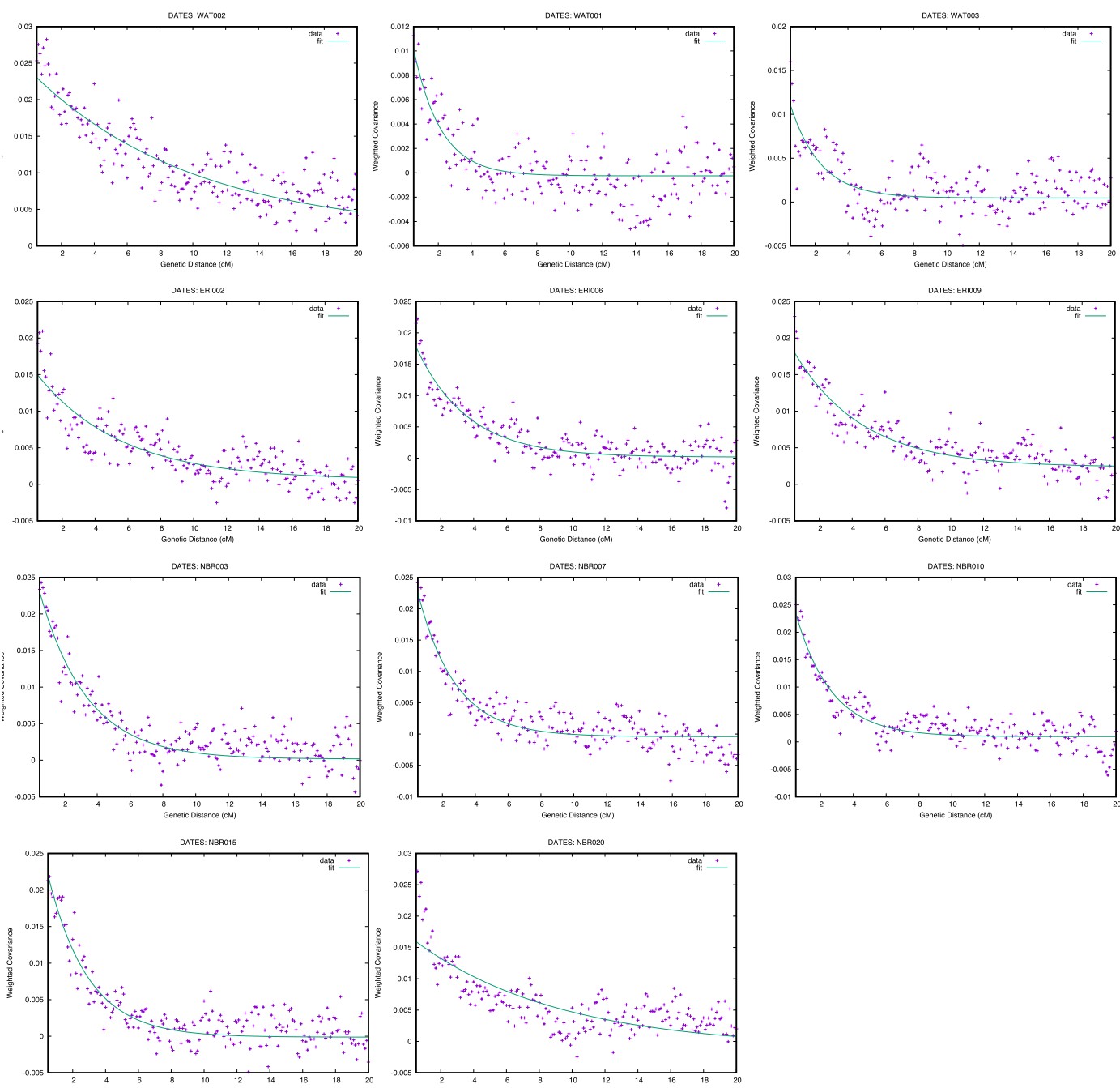

**Extended Data Fig. 4 | LD decay curves for the Admixture dating models.**

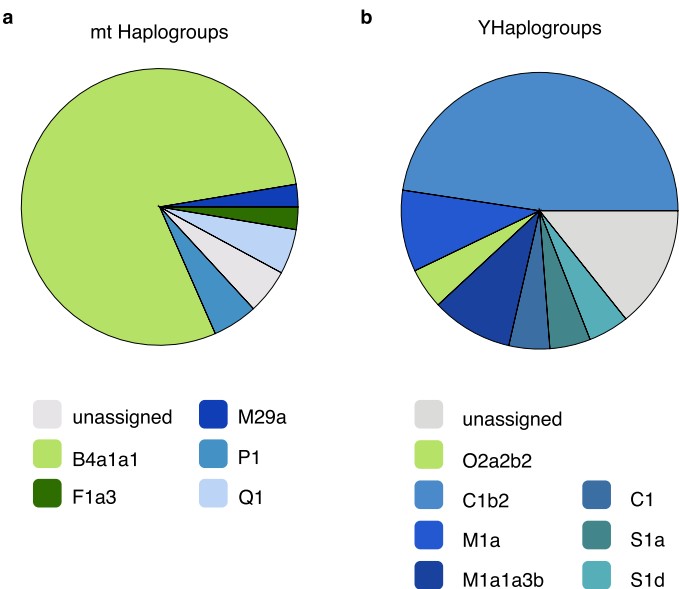

**a** mt Haplogroups  **b** YHaplogroups

**mt Haplogroups legend:**
- unassigned
- B4a1a1
- F1a3
- M29a
- P1
- Q1

**YHaplogroups legend:**
- unassigned
- O2a2b2
- C1b2
- M1a
- M1a1a3b
- C1
- S1a
- S1d

**Extended Data Fig. 5 | Uniparentally inherited markers.** Showing the LD decay between segments of Asian (Han.DG, Ami.DG, Atayal.DG, Igorot.DG, Kinh.DG, She.DG, Dai.DG) and Near Oceanic (Papuan.DG) ancestry in the individuals included in the analysis (Fig. 3d). (**a**) Mitochondrial haplogroups. (**b**) Y-chromosomal haplogroups.

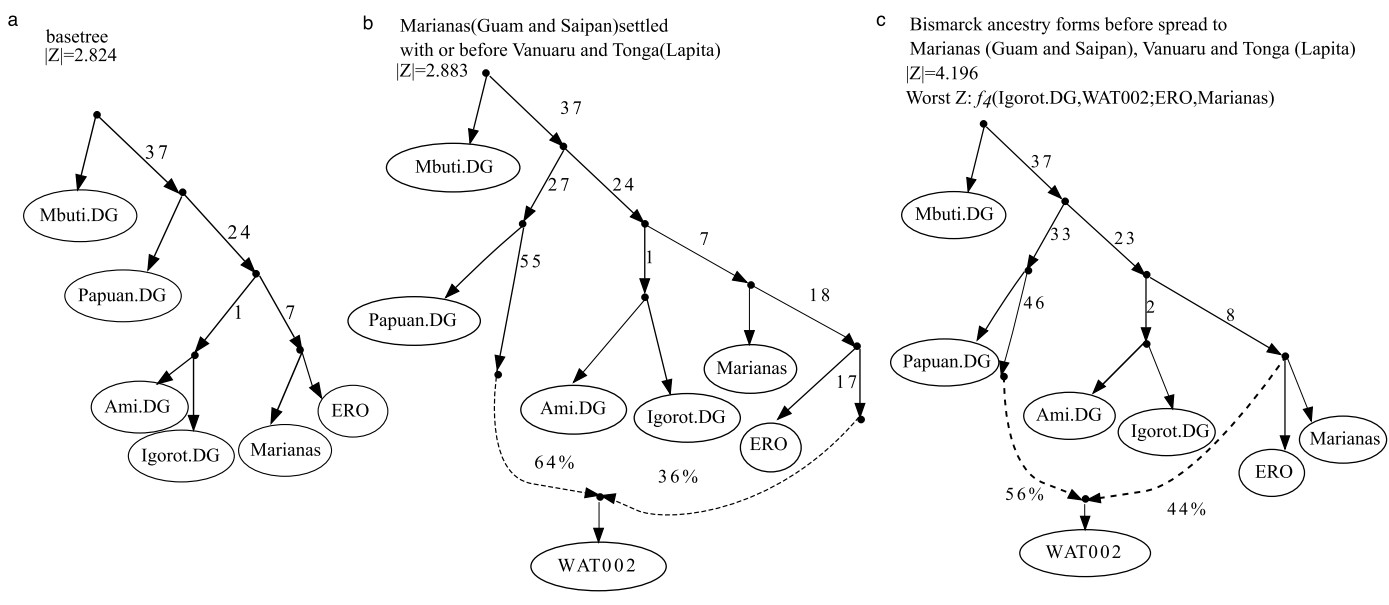

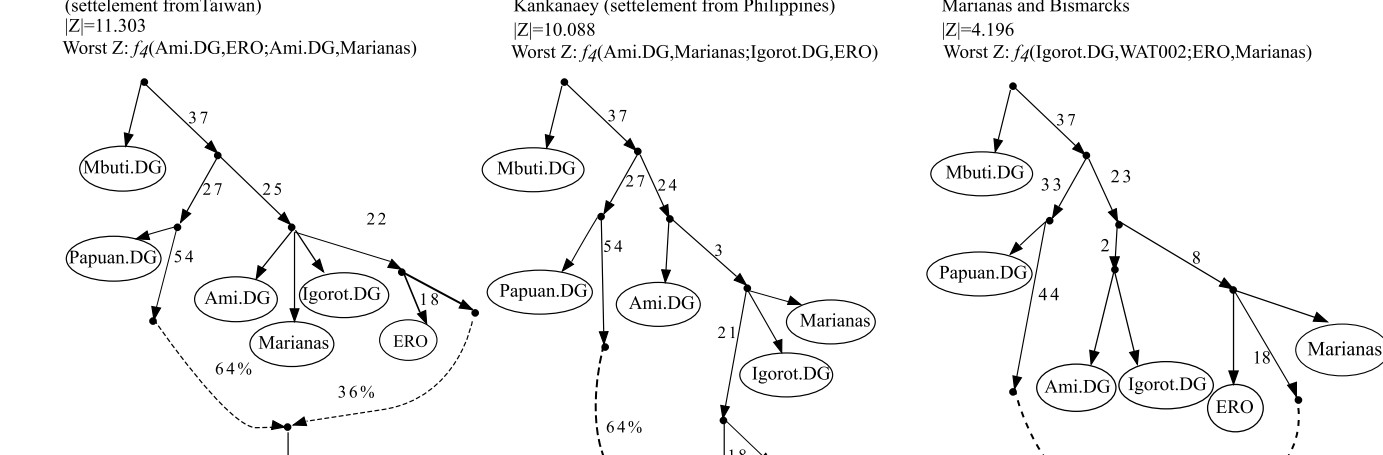

**Extended Data Fig. 6 | qpGraph model.** Testing relative affinities between the included populations. The constructed base tree (**a**) with the best tree modelling Amis and Kankanaey as sister groups, rather than a split of the Mariana Island ancestry from Amis (**d**) or Kankanaey (**e**); testing hypothesis i: settlement of the Mariana islands directly from Island South East Asia (**b**); and the alternative hypothesis ii: settlement of the Marianas after the settlement of the Bismarck Archipelago by Austronesian-related groups (**c**); for completness, model iii was tested, to exclude a back dispersal from Remote Oceania (**f**).

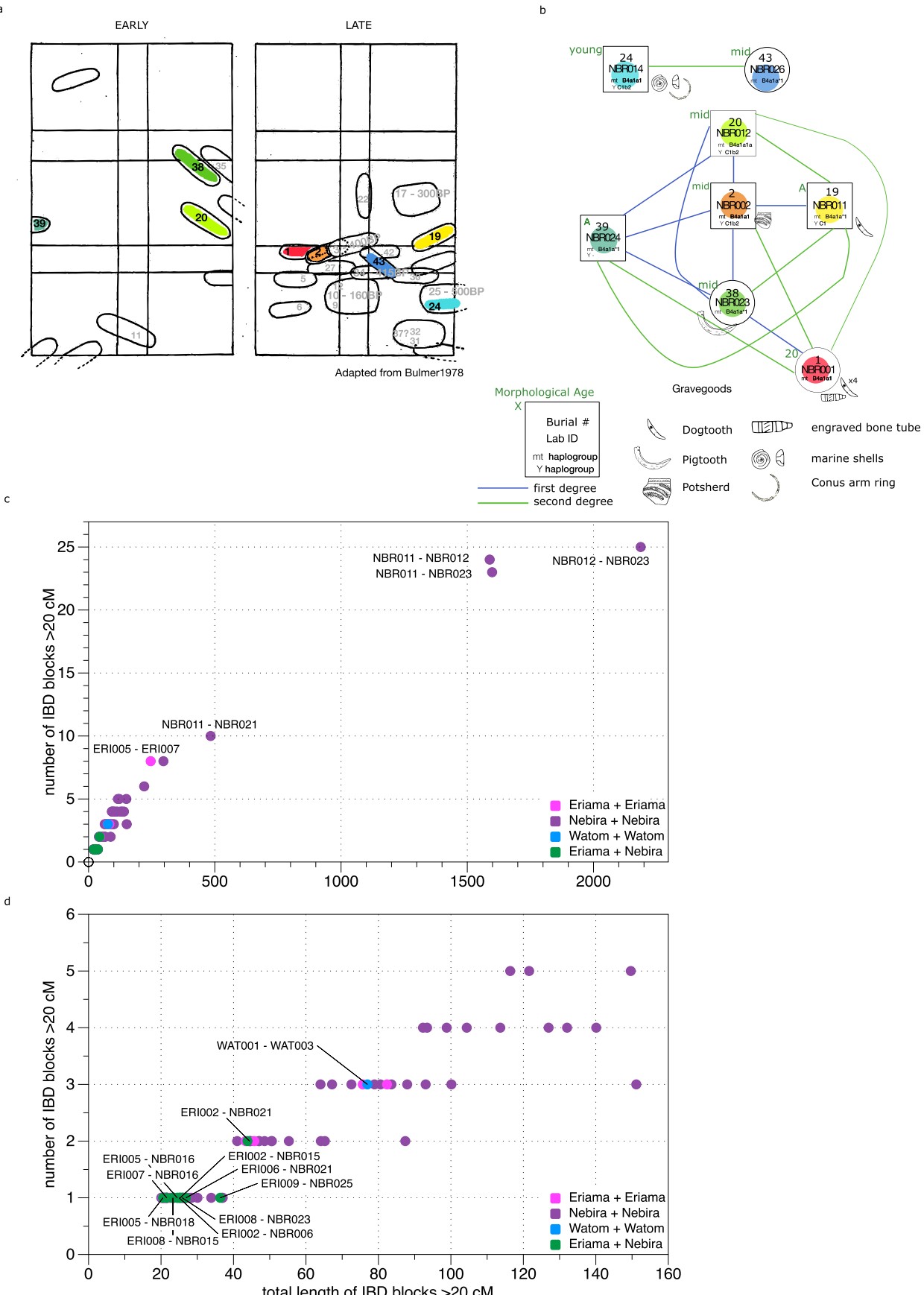

**Extended Data Fig. 7 | See next page for caption.**

**Extended Data Fig. 7 | Burial patterns and genetic relationships at the site of Nebira.** Position and orientation of graves from the early and late burial phase at Nebira (adapted from[66]) (**a**). Coloured graves indicate individuals with close genetic relationships. Individuals with black numbers indicate a close genetic relationship; individuals with grey numbers are included in the genetic analysis but do not indicate close genetic relations. Numbers correspond to burial numbers in[66]. Genetic relationship network (**b**), showing first (blue lines) and second degree (green lines) relationships, mitochondrial and Y-chromosomal haplogroups where available, morphological age and grave goods found with the individuals. (**c**) IDB block analysis for pairs of all individuals with successfully imputed genomes evaluating the total length of IBD blocks longer than 20 cM against the number of IBD blocks longer than 20 cm (**d**) and zoom in to the cluster of pairs with smaller sharing of blocks. Pairs are coloured by the pairing of sites as indicated in the legend. Pairs sharing IBD between the sites are labelled with the respective pair.

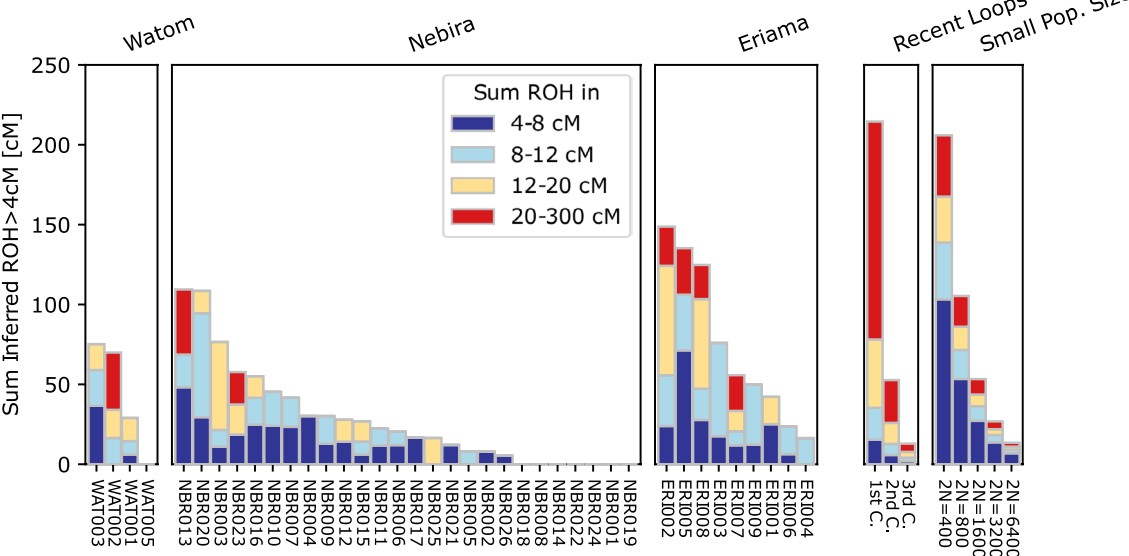

**Extended Data Fig. 8 | Sum of Runs of Homozygosity (ROH).** ROH of different sizes: 4–8 cm in dark blue, 8–12 cm in light blue, 12–20 cm in yellow and 20–300 cm in red.

# Reporting Summary

## Statistics

For all statistical analyses, confirm that the following items are present in the figure legend, table legend, main text, or Methods section.

| n/a | Confirmed | |
|---|---|---|
| ☐ | ☒ | The exact sample size (*n*) for each experimental group/condition, given as a discrete number and unit of measurement |
| ☐ | ☒ | A statement on whether measurements were taken from distinct samples or whether the same sample was measured repeatedly |
| ☐ | ☒ | The statistical test(s) used AND whether they are one- or two-sided<br>*Only common tests should be described solely by name; describe more complex techniques in the Methods section.* |
| ☐ | ☒ | A description of all covariates tested |
| ☐ | ☒ | A description of any assumptions or corrections, such as tests of normality and adjustment for multiple comparisons |
| ☐ | ☒ | A full description of the statistical parameters including central tendency (e.g. means) or other basic estimates (e.g. regression coefficient) AND variation (e.g. standard deviation) or associated estimates of uncertainty (e.g. confidence intervals) |
| ☐ | ☒ | For null hypothesis testing, the test statistic (e.g. *F*, *t*, *r*) with confidence intervals, effect sizes, degrees of freedom and *P* value noted<br>*Give P values as exact values whenever suitable.* |
| ☒ | ☐ | For Bayesian analysis, information on the choice of priors and Markov chain Monte Carlo settings |
| ☒ | ☐ | For hierarchical and complex designs, identification of the appropriate level for tests and full reporting of outcomes |
| ☒ | ☐ | Estimates of effect sizes (e.g. Cohen's *d*, Pearson's *r*), indicating how they were calculated |

*Our web collection on statistics for biologists contains articles on many of the points above.*

## Software and code

Policy information about availability of computer code

| Data collection | n/a |
|---|---|
| Data analysis | for details and citations see Supplementary Information. Clip&Merge, AdapterRemoval v.2, Burrows–Wheeler Aligner (BWA) v. 0.7.12, DeDup v. 0.12.2, SAMtools v. 1.3, pileupCaller v.8.6.5, DamageProfiler v0.3.1 , ANGSD v. 0.919 , Schmutzi, ADMIXTURE v.1.3.0 , smartpca v. 13050 , qp3Pop v. 5.0 , qpDstat v.5.0 , qpWave v. 410 , qpAdm v. 5.0 , DATES, nf-core/eager v. 2.4.0 , Nextflow v. 21.04.3, FastQC v. 0.11.9, AdapterRemoval v. 2.3.2, SAMtools version 1.12, Qualimap version 2.2.2-dev , MarkDuplicates version 2.26.0 , endorS.py version 0.4, DamageProfiler version 0.4.9, Geneious version 2019.2.3, MultiQC version 1.11 , HaploGrep version 2.4.0 , hapROH v.0.3a4 , Jupyter notebooks v.6.4.4, READ, ATLAS v.0.9 , GLIMPSE v.1.0.0, ancIBD v.0.2a2 , OxCal 4.4, https://github.com/hyl317/two_island |

For manuscripts utilizing custom algorithms or software that are central to the research but not yet described in published literature, software must be made available to editors and reviewers. We strongly encourage code deposition in a community repository (e.g. GitHub). See the Nature Portfolio guidelines for submitting code & software for further information.

## Data

Policy information about availability of data

All manuscripts must include a data availability statement. This statement should provide the following information, where applicable:

- Accession codes, unique identifiers, or web links for publicly available datasets
- A description of any restrictions on data availability
- For clinical datasets or third party data, please ensure that the statement adheres to our policy

The raw data of the captured libraries is available at the European Nucleotide Archive (ENA) https://www.ebi.ac.uk/PRJEB68153 under the accession number PRJEB68153. The genotypes of the newly published individuals can be sourced through the Poseidon framework community archive https://github.com/poseidon-framework/community-archive under 2024_NaegeleNatureEcologyEvolution.

The skeletal elements sampled at the Max-Planck-Institute of Geoanthropology in Jena, Germany, will be returned to the National Museum and Art Gallery of Papua New Guinea in Port Moresby (Nunguri and Tilu) and to the Department of Anatomy, University of Otago, Aotearoa (Nebira, Eriama and Watom) IN SPRING of 2025 to be curated with their respective skeletal assemblages.

## Research involving human participants, their data, or biological material

Policy information about studies with human participants or human data. See also policy information about sex, gender (identity/presentation), and sexual orientation and race, ethnicity and racism.

| | |
|---|---|
| Reporting on sex and gender | karyotypes of individuals were determined to inform X-and Y-chromosome specific analysis, i.e. sex-biased admixture and Y-chromosmal contamination, haplogroups. |
| Reporting on race, ethnicity, or other socially relevant groupings | n/a |
| Population characteristics | there were no co-variate relevant population characteristics |
| Recruitment | selection of individuals was determined by availability of archaeological material and preservation of DNA |
| Ethics oversight | Papua New Guinea National Art Gallery and Museum |

Note that full information on the approval of the study protocol must also be provided in the manuscript.

# Field-specific reporting

Please select the one below that is the best fit for your research. If you are not sure, read the appropriate sections before making your selection.

☒ Life sciences  ☐ Behavioural & social sciences  ☐ Ecological, evolutionary & environmental sciences

For a reference copy of the document with all sections, see nature.com/documents/nr-reporting-summary-flat.pdf

# Life sciences study design

All studies must disclose on these points even when the disclosure is negative.

| | |
|---|---|
| Sample size | was determined by preservation of DNA and number of individuals excavated and curated from a site |
| Data exclusions | one individual was excluded from the analysis based on a radiocarbon date overlapping with the present |
| Replication | the results were replicated within our team, but not shared with an external analyst for replication |
| Randomization | n/a |
| Blinding | n/a |

# Reporting for specific materials, systems and methods

We require information from authors about some types of materials, experimental systems and methods used in many studies. Here, indicate whether each material, system or method listed is relevant to your study. If you are not sure if a list item applies to your research, read the appropriate section before selecting a response.

## Materials & experimental systems

| n/a | Involved in the study |
|-----|----------------------|
| ⊠ ☐ | Antibodies |
| ⊠ ☐ | Eukaryotic cell lines |
| ☐ ⊠ | Palaeontology and archaeology |
| ⊠ ☐ | Animals and other organisms |
| ⊠ ☐ | Clinical data |
| ⊠ ☐ | Dual use research of concern |
| ⊠ ☐ | Plants |

## Methods

| n/a | Involved in the study |
|-----|----------------------|
| ⊠ ☐ | ChIP-seq |
| ⊠ ☐ | Flow cytometry |
| ⊠ ☐ | MRI-based neuroimaging |

# Palaeontology and Archaeology

| | |
|---|---|
| Specimen provenance | oversight of archaeological material, including human remains lies with the Papua New Guinea National Museum and Art Gallery. Permission for this study was granted to Hallie Buckley and Glenn Summerhayes. |
| Specimen deposition | The skeletal elements sampled at the Max-Planck-Institute of Geoanthropology in Jena, Germany, will be returned to the National Museum and Art Gallery of Papua New Guinea in Port Moresby (Nunguri and Tilu) and to the Department of Anatomy, University of Otago, Aotearoa (Nebira, Eriama and Watom) at the beginning of 2024 to be curated with their respective skeletal assemblages |
| Dating methods | AMS dates for this study were produced at the Curt-Engelhorn-Zentrum Archäometrie gGmbH in Mannheim, Germany. Collagen from bone and dentin was extracted using a modified Longin method and long molecules removed with ultrafiltration before freeze-drying the product. After the catalytic reduction to graphite the 14C content was measured with an AMS-System type MICADAS. The isotopic ratios of 14C/12C and 13C/12C of samples, standards (Oxalic acid II) and controls were measured simultaneously. The resulting 14C dates were normed with δ13C=-25‰ and calibrated using using Intcal20, OxCal 4.4, and marine20. The radiocarbon dates and quality collagen indicators (collagen yields, C/N ratios, %C and %N) are reported in Table S2 of the supplementary information |

⊠ Tick this box to confirm that the raw and calibrated dates are available in the paper or in Supplementary Information.

| | |
|---|---|
| Ethics oversight | Papua New Guinea National Museum and Art Gallery. |

Note that full information on the approval of the study protocol must also be provided in the manuscript.

# Plants

| | |
|---|---|
| Seed stocks | n/a |
| Novel plant genotypes | n/a |
| Authentication | n/a |

