## [Peer Review File · Nature Ecology & Evolution]

The impact of human dispersals and local interactions on the genetic diversity of coastal Papua New Gui

Corresponding Author: Dr Kathrin Naegele

Version 0:

Decision Letter:

10th January 2024

Dear Dr Naegele,

Your manuscript entitled "The impact of dispersals and local interactions on the ancient genetic diversity of Papua New Guinea over the past 2500 years." has now been seen by three reviewers, whose comments are attached. The reviewers have raised a number of concerns which will need to be addressed before we can offer publication in Nature Ecology & Evolution. We will therefore need to see your responses to the criticisms raised and to some editorial concerns, along with a revised manuscript, before we can reach a final decision regarding publication.

We therefore invite you to revise your manuscript taking into account all reviewer and editor comments. Please highlight all changes in the manuscript text file.

* If you have not done so already please begin to revise your manuscript so that it conforms to our Article format instructions at <http://www.nature.com/natecolevol/info/final-submission>. Refer also to any guidelines provided in this letter.

Link Redacted

Nature Ecology & Evolution is committed to improving transparency in authorship. As part of our efforts in this direction, we are now requesting that all authors identified as 'corresponding author' on published papers create and link their Open Researcher and Contributor Identifier (ORCID) with their account on the Manuscript Tracking System (MTS), prior to acceptance. ORCID helps the scientific community achieve unambiguous attribution of all scholarly contributions. You can create and link your ORCID from the home page of the MTS by clicking on 'Modify my Springer Nature account'. For more

information please visit please visit www.springernature.com/orcid.

Yours sincerely,

[redacted]

Reviewer expertise:

Reviewer #1: ancient DNA

Reviewer #2: Pacific and southeast Asian archaeology

Reviewer #3: ancient DNA, bioinformatics

Reviewers' comments:

Reviewer #1 (Remarks to the Author):

Nägele et al. present a new ancient DNA study of the genetic ancestry of Papua New Guinea and the Bismarck archipelago, including new archaeological information (radiocarbon dating and isotope analysis).

The authors sought to establish a time-transect that extends close to the initial settlement of the Lapita cultural complex in Near Oceania, employed commonly used tools in ancient DNA ancestry modelling (e.g., PCA, f-statistics, qpAdm, IBD block analysis) and a design strategy that reflects the extensive experience of this group of researchers doing very similar analysis in the past. Accordingly, I do not have any major comments on the strategy or the methods used in this work - which are standard and frequently replicated - only a few comments regarding the results.

1. Given the geographical proximity and synchronous sampling of the Nebira and Eriama sites, I share the authors surprise (lines 557-560) that the inferred history of the two populations, in particular the proportion of New Guinea ancestry, is so disparate.

If this is the case, then the most reasonable explanation, also suggested by the authors, is for the existence of a strong cultural barrier causing the observed population structure. In New Guinea, the most obvious cultural barrier would be language. The authors seem to think so as well (lines 561-571), but I found the sole paragraph on this possibility unsatisfactory. I thus think the manuscript would benefit from a more detailed comparison of the history of language evolution in this specific region, beyond the sparse information provided. Is the language barrier hypothesis testable? There is extensive literature on the languages of PNG that could perhaps be leveraged to test this hypothesis (or at least strengthen it). Given the multidisciplinary approach to this study, I think this could be an important addition.

However, could this observation instead be the result of an analytical artefact? I think it unlikely. But could the authors provide some evidence to rule this possibility out? For example, the aDNA analysis is based on captured data - how many 1240k sites have been included per individual, and do they overlap between the populations? Can you restrict the analysis to overlapping sites (even if needing to exclude low-coverage individuals and despite losing some power)? Such information would be more reassuring of the observations and rule out any potential ascertainment bias.

2. Figure 1: color codes are not clear. I assume they represent different populations, but cannot be sure. Please include an explanation in the figure legend.

3. Figure 2: Again, I am assuming that the color codes represent different populations. It is a bit challenging to examine the PCA plot, but it seems to me that there are some discrepancies regarding the number of individuals projected in the PCA and the numbers given in Figure 1. Specifically, there are 9 blue circles projected for Eriama (if I am observing correctly), but the authors refer only 8 individuals were analysed in Figure 1. In addition, I could not find the two Tilu individuals in the PCA. (The numbers for Nebira and Watom do seem correct).

Also, the statement in lines 255-258 should be rephrased. The fact that individuals cluster between two population groups (i.e., East Asians and Papuans) in the PCA does not mean they have 50% ancestry from each group, as it is now stated. In fact, the authors later show (qpAdm) that these proportions vary between the two populations.

4. Figure 3d: There seems to be a circle overlapping the filled block representing the admixture timing range for individual ERI007. Also, individuals ERI003, ERI006 and ERI009; and individuals ERI004 and ERI005 seem to have the exact same AMS dates (including range) - please double check.

5. Line 458: replace "Firgure 2" by "Figure 2"

6. Line 502: where is Figure 3e? I think the authors mean 3a.

Reviewer #2 (Remarks to the Author):

This paper details the implications of the first ever ancient DNA analyses from Papua New Guinea, a key location for understanding ancient migrations (c. 65,000 years ago into Sahul) and a staging post for later Oceanic migrations (of Austronesian peoples after c. 3500 years ago). The samples size is adequate, given the rarity of aDNA analyses in Oceania, though as the authors note only two samples have any bearing on the question of when 'East Asian' and 'Papuan' DNA admixed after Austronesian arrival in coastal New Guinea. Most of their samples are from very recent sites (the past 500 years) but nevertheless tell us something about admixture through time. Probably the study's most significant findings is that "the first, minimally admixed settlers remained isolated from local populations with Papuan-related ancestry, despite geographical proximity". However, the term "isolated" here is very problematic - they may have shared close relationships (e.g., exchange) but not intermarried. Which is very interesting indeed.

The data and supplementary information are of a high quality, and there are no obvious errors. I must caution that aDNA is not my field, so my comments are on the narrative and the archaeology (including the radiocarbon dating).

I am slightly concerned about the vagueness of the term East Asian – I can tell from the Supplementary Information that the authors have given thought to terminology. Is there an alternative that cannot be confused with modern sociopolitical regions? Could "Indigenous East Asian" be used, given Indigenous Taiwanese aDNA samples were used as a comparison? Or Austronesian? (This has its own problems, I am aware). On a related note, the relationship between Austronesian dispersals and East Asian ancestry needs to be mentioned in the Abstract and early in the Introduction.

In my opinion, the Title and Abstract need rewriting to show readers the headline of your findings. "The impact" is a bit vague. The Abstract too needs to convey your main findings in more plain language, and there is currently a very awkward transition between the 2700-2100 results and the 'Three coastal sites' which sounds like an entirely new Abstract/paper is starting on line 69. This needs rewriting to flow better.

One area the paper needs sharpening is in its discussion of the Papua Hiccup. Given that most of the samples are from the past 500 years, and admixture is inferred not long before that, much of the actual data relates to a period just prior to the emergence of vast ethnographically known exchange routes (e.g., Hiri). However, the authors have incorrectly cite Shaw and Allen for this period, when first mention was made by Geoff Irwin and Jim Rhoads – since this time the most extensive discussion of the period is in Skelly and David, and only Jim Allen attributes the abandonment of coastal sites to "challenging environmental fluctuations" – the authors should expand their discussion upon reading the literature on this period more closely (add nuance). It is not surprising that there is a link between the genetics, the development of the hiri, and the development of Motu-Koita interrelations, and this could perhaps be explained more clearly also.

Small notes:

Line 56. What are ancient inhabitants? Why not just inhabitants?

Line 83. And the Aru Islands?

Line 142. Cite original papers for Papuan/Ceramic Hiccup

Line 152. We need to get a sense of your argument in this part of the introduction

Line 170. Should be 'Affect how'

Line 354. What are 'seafaring probabilities'?

Line 375. Why is Bulmer not cited?

Line 383. This is a major issue for me and needs fixing. Of course 'family was not necessarily restricted to genetic relatedness' – any familiarity with the Indigenous societal structures of PNG would make this very clear, and some nuance needs to be shown here. Western genetics is not a holistic analysis of 'family' or 'kin' – these are socially constructed.

Line 505. This inferred date is inferred, and probably due to your sample size. Suggest re-emphasising the limited dataset. It is highly improbable that admixture took more than 1000 years

Line 519-521. This alternative explanation is completely unsupported by the archaeological evidence. Despite a great deal of debate back in 2012 about whether Lapita people ever visited any mainland, the entire Lapita community accepts there were settlements – if they had only found pottery, this would be a reasonable statement.

Line 539. Rhoads and Urwin need citing for hiri emergence - Shaw et al for the Kula.

Line 550. Reflect here on the emergence of Motu-Koita relationships/kinship.

Line 605-606. For me, this is one of the headlines of your research. But I don't think it is well enough foregrounded throughout. Make this pop out to readers.

Line 643-645. Does Alois Kuaso not reach the input threshold for authorship? It seems odd to acknowledge input with no authorship.

Reviewer #3 (Remarks to the Author):

The authors present a manuscript on the recent population history of coastal New Guinea, using ancient DNA from the last 2500 years. This is the first ancient genome data reported from this important but somewhat neglected part of the world, and thus represents a valuable contribution. DNA preservation is presumably not very good in this region, and so the success in generating data is an impressive achievement. The authors then use this data, together with isotope and dental calculus data, to study the population history of the region. The paper provides quite extensive background and archaeological context on the Austronesian and Lapita expansions, and is overall well-written. The analyses are standard and appear well-executed. I think some clarifications are needed in terms of how analyses, public data, figures/tables etc. are described, as I elaborate on at various points below. I might have expected some more results on within-PNG structure and changes in addition to the focus on the Austronesian expansion and East Asian admixture, but it is of course up to the authors to choose what aspects to focus on.

I think it would be useful to report in the abstract the number of individuals from which genomic data was recovered – this is useful for readers wanting to get a quick overview.

The results obtained on the East Asian admixture question are interesting, but not necessarily very ground-breaking. Most (all?) of the samples substantially post-date the arrival of Austronesians to coastal PNG, so it's not surprising that they do have East Asian admixture or that they have it in variable amounts (just like it's variable today in PNG). The most interesting part of these results is the admixture timing (it would be useful if the sentence "performed an admixture dating analysis" also mentioned what method/approach was used, e.g. at the very least linkage disequilibrium). The result interestingly indicates admixture occurred not until 1,000 years after Austronesian arrival, suggesting a quite long period without admixture – though caveats apply, which the authors discuss appropriately.

I could not find any description of what modern datasets were used in this study. This must be described, and source publications cited. For example, there is reference to data from the present-day group Tolai, but it's unclear what this data is – whole-genome data, array data, how many individuals, from what publication? Population labels like "Denisova_published.DG" are not particularly useful – a few people will know where this data derives from, but this is not sufficient information to enable other researchers to understand what was done and to replicate the analyses.

The IBD and ROH analyses are somewhat interesting I suppose, but the results are not so easy to interpret when there is no present-day context. It would have been natural to ask how the results in the ancient genomes compare to present ROH levels in PNG, but the authors do not do this (there is quite a bit of modern data from PNG available from previous studies). Though I would not necessarily demand that the authors expand the analysis in this way, as it would be quite a lot of work.

I can't quite get my head around Figure 2 – some samples appear to be missing? I.e. I can't locate any symbols corresponding to present-day PNG, Australia, Bismarck or Solomons. There are text labels for these groups annotated onto the plot, but seemingly no points around those labels? Perhaps a figure processing error.

Supplementary Figure 2 is extremely low-resolution and completely illegible, so the claims made in relation to these results cannot be reviewed currently.

Line 265 onwards describes how two individuals "were consistent with exclusively Papuan-related ancestry", but line 278 says "Having established that none of the individuals derived from only one stream of ancestry". These statements do not seem compatible.

There is confusion with regards to the numbering of supplementary tables: the numbers listed in the supplements PDF, the numbers on the Excel file sheets, and the numbers on the first lines of each table, do not agree with each other.

Supplementary Table 12/7 qpWave/qpAdm. It's a bit tricky to understand what the various tables mean. In particular the first table, which has "ID", "Group", and "p-value" as column headings – I can't figure out what this table is. Also, more information is needed on the population "New_Guinea" used in the reference community – where precisely in New Guinea this is from is of course highly relevant in this context. Some fields are colour-coded (red, grey), but what this means is not explained. I would recommend against colour-coding as a means to convey information, as this is not machine-readable.

Admixture graph testing for the peopling of the Marianas islands: as there are no suitable ancient genomes from the Philippines, the authors rather leave that possible ancestry unmodelled in the graphs. But should it not be fine to use some present-day Austronesian group from the Philippines, e.g. Kankanaey/Igorot, who seemingly do not have any Papuan-related ancestry (as done in Skoglund et al. 2016). It seems like that would help constrain things, and then hopefully allow a

firmer conclusion on this question?

Data availability: The ENA accession number appears inactive, so the archived data cannot be evaluated at this point. It's important that all of the reads, not just those that mapped to the reference genome, are archived (see: <https://www.biorxiv.org/content/10.1101/2023.05.15.540553v1>).

*****END*****

Version 1:

Decision Letter:

11th June 2024

Dear Dr Naegele,

Your Article, "The impact of human dispersals and local interactions on the ancient genetic diversity of Papua New Guinea over the past 2500 years." has now been seen by the original reviewers. You will see from their comments copied below that while they find your work of considerable potential interest, they have raised quite substantial concerns that must be addressed. In light of these comments, we cannot accept the manuscript for publication, but would be very interested in considering a revised version that addresses these serious concerns.

We hope you will find the reviewers' comments useful as you decide how to proceed. If you wish to submit a substantially revised manuscript, please bear in mind that we will be reluctant to approach the reviewers again in the absence of major revisions.

While the reviewers don't identify any new issues with the revised manuscript, two of the reviewers feel that efforts to address first round comments leave something to be desired. In particular, reviewer 1 takes up reviewer 3's previous comment about comparing with present day or more recent PNG genetic data, and expresses disappointment that you don't feel able to conduct linguistic analyses to explore potential cultural correlates of the genetic data patterns. While we accept your argument that this linguistic analysis may be beyond the scope of the current paper, it's apparent from reviewer 2's report that the current cultural interpretations need more thorough contextualisation and thought and these should be thoroughly integrated into the manuscript making use of the existing literature. We also encourage you to follow the original suggestions to compare with what's known about the genetic history of the region.

If you choose to revise your manuscript taking into account all reviewer and editor comments, please highlight all changes in the manuscript text file [OPTIONAL: in Microsoft Word format].

* Include a "Response to reviewers" document detailing, point-by-point, how you addressed each referee comment. If no action was taken to address a point, you must provide a compelling argument. This response will be sent back to the referees along with the revised manuscript.

* If you have not done so already we suggest that you begin to revise your manuscript so that it conforms to our Article format instructions at <http://www.nature.com/natecolevol/info/final-submission>. Refer also to any guidelines provided in this letter.

Link Redacted

If you wish to submit a suitably revised manuscript we would hope to receive it within 6 months. If you cannot send it within this time, please let us know. We will be happy to consider your revision so long as nothing similar has been accepted for publication at Nature Ecology & Evolution or published elsewhere.

Nature Ecology & Evolution is committed to improving transparency in authorship. As part of our efforts in this direction, we are now requesting that all authors identified as 'corresponding author' on published papers create and link their Open Researcher and Contributor Identifier (ORCID) with their account on the Manuscript Tracking System (MTS), prior to acceptance. This applies to primary research papers only. ORCID helps the scientific community achieve unambiguous attribution of all scholarly contributions. You can create and link your ORCID from the home page of the MTS by clicking on 'Modify my Springer Nature account'. For more information please visit www.springernature.com/orcid.

Thank you for the opportunity to review your work.

Yours sincerely,

[redacted]

Reviewer expertise:

as before

Reviewers' comments:

Reviewer #1 (Remarks to the Author):

I have now completed my second round of review of the manuscript submitted by Nägele et al. I have no further comments to the authors beyond the initial suggestions/concerns raised after reviewing the first version of this work.

I will, however, express my disappointment in the authors' seeming reluctance to try and correlate their results to linguistic diversity or the genetic history of the region (as suggested by reviewer 3 on multiple instances). In particular, I found their arguments for not performing further analyses, which would clearly strengthen the manuscript, to be very weak. This necessarily leaves me wondering if the reason for their resistance is instead more related to the amount of work (and time) necessary to implement them, or the fact that such analyses are not typical of ancient human DNA studies and thus not as trivial to implement in a standardised/industrial fashion.

I thus find the revised version of the manuscript unsatisfactory, as some of the main comments raised by the reviewers were not properly addressed. The authors did an exceptional job correcting the numerous errors identified in Figures, legends, or nomenclature, but showed an odd reluctance to include additional hypotheses-testing. I still find the manuscript to be solid work, and thus suitable for publication in Nature Ecology and Evolution. However, my recommendation will necessarily be to urge the authors to perform the requested analyses before publication.

Reviewer #2 (Remarks to the Author):

Overall, I am highly satisfied with the responses of the author to all of the reviewers' critiques (but see a couple of specific remaining problems below). They have engaged thoughtfully with issues of terminology and how their results are narrated for a broad scientific readership.

There remain some grammatic errors in the manuscript, but I presume Nature will work with the authors on this in the next stage of publication.

For example, in the Abstract (which is much more readable, and conveys better the significance of the research) it says, "The individuals span 2500 years of human habitation and highlight the influences". But no individual spans 2500 years, and no individual highlights anything (rather, their results and interpretation do the highlighting, and their oldest sample is from 2500 years ago). This is a problem of expression, and it is not isolated.

Despite saying they have amended the references for the Hiri and Papuan Hiccup, they now erroneously cite Skelly and David 2017 for the (unproven) idea that there was a breakdown in marine resources caused by climatic changes (Line 522). Skelly and David emphasise social transformations including the possibility that increased warfare led to communities relocating inland to defensive settlements (see Rhoads also!). Where is this mentioned in the manuscript? Only some quite vague climatic evidence is invoked instead.

The authors still need to make clear that Rhoads (1982) developed the concept of the Papuan Hiccup & Irwin first developed the Ceramic Hiccup. They should also briefly mention the various social, economic, etc. explanations that have been put forward for this era beyond climate (Allen's 2010 paper is not the most up to date work on this). The final two chapters of Skelly and David's book will be key for dealing with explanations for the post-650 BP era as it is the most comprehensive

review of arguments about this period.

It also seems odd that the authors have introduced a new and unqualified reference to a 'climatically challenging' period immediately prior to 650 BP in the Abstract. Previously the authors more tentatively stated in the main text that an improving climate 'possibly' enabled the post-650 BP transformations. Where are the independent climate records to back this 'challenging period' for PNG's south coast? What do the cores at Lake Kutubu tell us?

I know the authors have limited words to spend, but it is important that they convey that there is no consensus on what drove the post-650 BP re-emergence of coastal settlements and long-distance exchange networks.

Reviewer #3 (Remarks to the Author):

The authors have responded well to the reviewer comments, and clarified various things in figures and tables. I have no further comments, and would recommend the paper for publication. I find the Discussion a bit overly long personally, but it is up to the authors to decide. I just leave a couple of final minor things below.

Figure 3a: It needs to be indicated, at least in the figure legend, what the numbers on the right hand-side are (presumably these are p-values). Also, the numbers currently overlap the figure. It also needs to be indicated what the error bars represent.

"an excess of Austronesian-related ancestry on the X-chromosome ranging from 10 to 60 percent" – should this rather say "percentage points", instead of "percent"?

Supplementary Figure S2 is very low-resolution, it's virtually impossible to read the population labels (they also seem to be upside down?).

Version 2:

Decision Letter:

17th December 2024

Dear Kathrin,

Thank you for submitting your revised manuscript "The impact of human dispersals and local interactions on the genetic diversity of coastal Papua New Guinea over the past 2500 years." (NATECOLEVOL-23112683B). It has now been seen again by two of the original reviewers and their comments are below (note that reviewer 1 had no comments for the authors, just to the editor). The reviewers find that the paper has improved in revision, and therefore we'll be happy in principle to publish it in Nature Ecology & Evolution, pending minor revisions to satisfy the reviewers' final requests and to comply with our editorial and formatting guidelines.

We are now performing detailed checks on your paper and will send you a checklist detailing our editorial and formatting requirements. Please do not upload the final materials and make any revisions until you receive this additional information from us. Note that although we would normally try to get this checklist to you within a week, given the impending Christmas/New Year holidays it is likely to be early January, I'm afraid.

Sincerely,

[redacted]

Reviewer #1 (Remarks to the Author):

NA

Reviewer #2 (Remarks to the Author):

My remaining concerns (2nd round of reviews) were about the quality of the archaeological literature review. If the paper had been published in its previous state, scholars working on the south coast of PNG would have read this paper and thought (possibly correctly) that the authors had not actually read the literature describing the emergence of exchange systems

(especially the literature from the Papuan Gulf).

The difference between that manuscript and this one is night and day. The authors have produced a highly nuanced and evidence-based discussion that reflects the current state of research and positions their own contribution much more effectively.

I suggest that the authors add a small caveat next to their citation of the Medieval Warm Period, simply stating that the IPCC (<https://www.ipcc.ch/>) is not convinced there is enough evidence for a global (southern hemisphere) climatic era. The tree ring data from Australia actually contradicts this period. There is some positive evidence from Pacific coral (<https://www.science.org/doi/10.1126/science.1240837>) but the authors should very briefly flag that more data is needed (i.e., their paper is important for highlighting the need for future research on the social and climatic drivers for exchange system transformations, etc).

Excellent rewrite, still an unfortunate case of asking external reviewers rather than co-authors to do a fair bit of literature review legwork.

Response to the reviewers

We thank all reviewers for their time and effort in assessing our study. We have carefully considered the points raised and provide the responses below marked in bold font.

Reviewer #1 (Remarks to the Author):

Nägele et al. present a new ancient DNA study of the genetic ancestry of Papua New Guinea and the Bismarck archipelago, including new archaeological information (radiocarbon dating and isotope analysis).

The authors sought to establish a time-transect that extends close to the initial settlement of the Lapita cultural complex in Near Oceania, employed commonly used tools in ancient DNA ancestry modelling (e.g., PCA, f-statistics, qpAdm, IBD block analysis) and a design strategy that reflects the extensive experience of this group of researchers doing very similar analysis in the past. Accordingly, I do not have any major comments on the strategy or the methods used in this work - which are standard and frequently replicated - only a few comments regarding the results.

1. Given the geographical proximity and synchronous sampling of the Nebira and Eriama sites, I share the authors surprise (lines 557-560) that the inferred history of the two populations, in particular the proportion of New Guinea ancestry, is so disparate.

If this is the case, then the most reasonable explanation, also suggested by the authors, is for the existence of a strong cultural barrier causing the observed population structure. In New Guinea, the most obvious cultural barrier would be language. The authors seem to think so as well (lines 561-571), but I found the sole paragraph on this possibility unsatisfactory. I thus think the manuscript would benefit from a more detailed comparison of the history of language evolution in this specific region, beyond the sparse information provided. Is the language barrier hypothesis testable? There is extensive literature on the languages of PNG that could perhaps be leveraged to test this hypothesis (or at least strengthen it). Given the multidisciplinary approach to this study, I think this could be an important addition.

We thank the reviewer for their agreement. As we discuss in the manuscript, language can be cultural barrier between communities, however, in situations where many languages exist side-by-side in a small area, lots of language mixing can be observed. This is particularly the case in the Western Pacific and generally in New Guinea. This observation is a product of cultural mixing and intense/prolonged social interaction both so much so historically that they create obstacles for historical linguists in classifying some languages. With this regard, we chose to not go into depth discussing or testing this in our manuscript, as the historical impacts would lead to possibly wrong conclusions when putting together with the ancient DNA data. Linguistic diversity could be a cultural barrier in these particular areas of New Guinea, and it would be in principle possible to test this, however this would be a much larger separate study.

However, could this observation instead be the result of an analytical artefact? I think it unlikely. But could the authors provide some evidence to rule this possibility out? For example, the aDNA analysis is based on captured data - how many 1240k sites have been included per individual, and do they overlap between the populations? Can you restrict the analysis to overlapping sites (even if needing to exclude low-coverage individuals and despite losing some power)? Such information would be more reassuring of the observations and rule out any potential ascertainment bias.

The individual SNP coverage for the two sites is (for tropical environments) surprisingly good, with an average coverage of >430 000 sites for Nebira and > 640 000 for Eriama, ranging between 11 000 to 610 000 and 430 000 to 790 000 respectively (see table S1 for an overview). The number of SNPs overlapping the two should be >103 000 SNPs, as indicated as the overlap used to calculate the f4 statistic Mbuti.DG, Papuan.DG; Eriama, Nebira, added to the Supplementary data Table S11 (Papuan related ancestry section

column I, line 5). This test has a strongly negative result, showing the excess affinity of Eriama to Papuan, when compared to Nebira. We would therefore argue that a bias based on the coverage can be excluded, and we do observe a true signal, consistent with qpAdm, Admixture and PCA.

2. Figure 1: color codes are not clear. I assume they represent different populations, but cannot be sure. Please include an explanation in the figure legend.

The different colours represent the different archaeological sites/complexes and are used to represent individuals from these sites in the PCA. We have clarified this by adding the statement to the figure caption.

3. Figure 2: Again, I am assuming that the color codes represent different populations. It is a bit challenging to examine the PCA plot, but it seems to me that there are some discrepancies regarding the number of individuals projected in the PCA and the numbers given in Figure 1. Specifically, there are 9 blue circles projected for Eriama (if I am observing correctly), but the authors refer only 8 individuals were analysed in Figure 1. In addition, I could not find the two Tilu individuals in the PCA. (The numbers for Nebira and Watom do seem correct).

Thank you for pointing this out. The mistake in the plotting was corrected. Tilu is now shown as a square, TIL004 is transparent because of the low number of SNPs, as are NBR008 and WAT006.

Also, the statement in lines 255-258 should be rephrased. The fact that individuals cluster between two population groups (i.e., East Asians and Papuans) in the PCA does not mean they have 50% ancestry from each group, as it is now stated. In fact, the authors later show (qpAdm) that these proportions vary between the two populations.

We agree with the reviewer that the placement does not reflect a 50% percent ancestry from each group, and realize that the phrase, although not intended to state that, did suggest so. We have now changed the wording to state the intended conclusion that “*The placement of the Eriama and Nebira individuals suggests a genetic composition derived from a mixture of Papuan-related ancestry most similar to present-day highland populations of New Guinea (Supplementary Fig. 2) and of East Asian-related ancestryA.*”

4. Figure 3d: There seems to be a circle overlapping the filled block representing the admixture timing range for individual ERI007. Also, individuals ERI003, ERI006 and ERI009; and individuals ERI004 and ERI005 seem to have the exact same AMS dates (including range) - please double check.

Thank you for pointing out another plotting artifact, the overlapping dot has been removed. For a maximum conservative approach, we show the date ranges of the calibrated dates BP 95% probability range, as reported in extended data Table S2. Despite slightly different uncalibrated dates, the calibration does equalize the dates reported for the respective individuals. For increased transparency, we have added the details on the reported dates to the Figure 3d caption.

5. Line 458: replace "Firgure 2" by "Figure 2". **Changed.**

6. Line 502: where is Figure 3e? I think the authors mean 3a. **Correct, thank you for pointing that out.**

Reviewer #2 (Remarks to the Author):

This paper details the implications of the first ever ancient DNA analyses from Papua New Guinea, a key location for understanding ancient migrations (c. 65,000 years ago into Sahul) and a staging post for later Oceanic migrations (of Austronesian peoples after c. 3500 years ago). The samples size is adequate, given the rarity of aDNA analyses in Oceania, though as the authors note only two samples have any bearing on the question of when ‘East Asian’ and ‘Papuan’ DNA admixed after Austronesian arrival in coastal New Guinea.

Most of their samples are from very recent sites (the past 500 years) but nevertheless tell us something about admixture through time. Probably the study's most significant findings is that "the first, minimally admixed settlers remained isolated from local populations with Papuan-related ancestry, despite geographical proximity". However, the term "isolated" here is very problematic - they may have shared close relationships (e.g., exchange) but not intermarried. Which is very interesting indeed.

We thank the reviewer for their caution regarding terminology used and agree – “isolation” is more than just the absence of genetic mixing. To add nuance, we have changed the wording to “genetic isolation”.

The data and supplementary information are of a high quality, and there are no obvious errors. I must caution that aDNA is not my field, so my comments are on the narrative and the archaeology (including the radiocarbon dating).

I am slightly concerned about the vagueness of the term East Asian – I can tell from the Supplementary Information that the authors have given thought to terminology. Is there an alternative that cannot be confused with modern sociopolitical regions? Could “Indigenous East Asian” be used, given Indigenous Taiwanese aDNA samples were used as a comparison? Or Austronesian? (This has its own problems, I am aware).

On a related note, the relationship between Austronesian dispersals and East Asian ancestry needs to be mentioned in the Abstract and early in the Introduction.

Thank you for regarding the terminology. We have also given thought to the term for the broad genetic signal observed, and decided here to use a broader assignment of “East Asian”. Considering “Indigenous East Asian” when mostly Taiwanese aboriginal genomes were used for the modeling, disregards other indigenous East Asian groups. At the same time, we wanted to avoid using “aboriginal Taiwanese” as a term for a genomic signal preceding the naming of the island as Taiwan (relating to your own concerns of present-day sociopolitical regions). Considering the use of “Austronesian” for the genetic signal, we also decided to avoid this language-based naming because Austronesian speaking groups today are very variable in their genetic ancestry, the cultural representation and geographic extent of the language. Similar to the oversimplification of the diverse Papuan ancestry profiles to “Papuan-related”, we decided to default to a broad term of “East Asian-related”, not inspired by sociopolitical regions, but by the geographical region that is most likely the ancestral home of the genetic profile dispersing in East-, South East- and Island South East Asia ~7000 – 5000 years ago. We have added a paragraph to the “Terminology” section of the Supplement to explain the terminology:

“A similar problem arises when describing the general genetic profile deriving from a dispersal from Asia. In this study, we refer to the genetic component associated with a dispersal from the mainland and island South East Asia as “East Asian-related”, similar to the oversimplification of the diverse Papuan ancestry profiles to “Papuan-related. This term should not be understood as a reflection of a sociopolitical region, but inspired by the broad geographical region that is most likely the ancestral home of the genetic profile dispersing in East-, South East- and Island South East Asia ~ 7000 -500 years ago. Alternative terms could be “Indigenous East Asian”. However, since mostly Taiwanese and Philippine indigenous genomes were used for or the modeling this would disregard the many indigenous East Asian groups with different genetic profiles. Considering the use of “Austronesian” for the genetic signal, we also decided to avoid this language-based naming because Austronesian speaking groups today are very variable in their genetic ancestry, the cultural representation and geographic extent of the language.”

In my opinion, the Title and Abstract need rewriting to show readers the headline of your findings. “The

impact” is a bit vague. The Abstract too needs to convey your main findings in more plain language, and there is currently a very awkward transition between the 2700-2100 results and the ‘Three coastal sites’ which sounds like an entirely new Abstract/paper is starting on line 69. This needs rewriting to flow better.

Thank you for this suggestion. We have rewritten the abstract and hope that it now conveys the main findings more efficiently regardless of the limit of 200 words imposed by the journal.

One area the paper needs sharpening is in its discussion of the Papuan Hiccup. Given that most of the samples are from the past 500 years, and admixture is inferred not long before that, much of the actual data relates to a period just prior to the emergence of vast ethnographically known exchange routes (e.g., Hiri). However, the authors have incorrectly cite Shaw and Allen for this period, when first mention was made by Geoff Irwin and Jim Rhoads – since this time the most extensive discussion of the period is in Skelly and David, and only Jim Allen attributes the abandonment of coastal sites to “challenging environmental fluctuations” – the authors should expand their discussion upon reading the literature on this period more closely (add nuance). It is not surprising that there is a link between the genetics, the development of the hiri, and the development of Motu-Koita interrelations, and this could perhaps be explained more clearly also.

Thank you for suggesting additional literature. We have carefully considered the suggested readings and hope to have included them to your satisfaction, some of those edits detailed in the suggestions below.

Small notes:

Line 56. What are ancient inhabitants? Why not just inhabitants?

The term ancient inhabitants was used to contrast the present-day inhabitants, whose genomic information is available to some extent. However, we see how this wording could be understood as excluding people today from an impact in the region, and have therefore removed “ancient” from this sentence.

Line 83. And the Aru Islands? **Added to the sentence**

Line 142. Cite original papers for Papuan/Ceramic Hiccup

We thank the reviewer again for the suggested literature and have now updated the citation and expanded the explanation of the phenomenon.

Line 152. We need to get a sense of your argument in this part of the introduction

Line 170. Should be ‘Affect how’

Line 354. What are ‘seafaring probabilities’?

We thank you for pointing out the use of jargon here and have replaced the term with “highest probabilities of landfall in the Marianas during a seavoyage”

Line 375. Why is Bulmer not cited?

Is now cited, as should have been.

Line 383. This is a major issue for me and needs fixing. Of course ‘family was not necessarily restricted to genetic relatedness’ – any familiarity with the Indigenous societal structures of PNG would make this very clear, and some nuance needs to be shown here. Western genetics is not a holistic analysis of ‘family’ or ‘kin’ – these are socially constructed.

Thank you for this comment. We completely agree with the notion that genetic cannot singlehandedly provide insight into family and kin. However, combined with other lines of evidence, such as consumption patterns, distribution of burial goods and burial customs, it can help interpreting the archaeological context. In the current study, unfortunately none of the patterns observed would allow us to delve into such questions and compare the patterns with those observed in present-day communities where we know how kin and family are defined. The statement that “family is not necessarily restricted to genetic relatedness” is addressed at a readership that might not be familiar with the breadth of definitions of kin in various cultures and parts of the world.

Line 505. This inferred date is inferred, and probably due to your sample size. Suggest re-emphasising the limited dataset. It is highly improbable that admixture took more than 1000 years

We do share your skepticism regarding the significant delay between the arrival of Lapita pottery makers and the detection of the genetic admixture. From the genetic line of evidence we are currently unable to look past the admixture that occurred 1000 years ago. Hopefully the investigation of a time transect in the region will help to improve the inference (exemplified in Posth, Nägele et al. 2018). Until then, we will have to report the date of 1000 BP, of course pointing to the limitations, which we hope we were able to convey better in this version, regardless of the significant cuts to meet the word limit.

Line 519-521. This alternative explanation is completely unsupported by the archaeological evidence. Despite a great deal of debate back in 2012 about whether Lapita people ever visited any mainland, the entire Lapita community accepts there were settlements – if they had only found pottery, this would be a reasonable statement.

Thank you for raising this point, we agree and have removed this statement.

Line 539. Rhoads and Urwin need citing for hiri emergence - Shaw et al for the Kula.
We have updated the citations.

Line 550. Reflect here on the emergence of Motu-Koita relationships/kinship.
We have included a statement relating the observed pattern to the Motu-Koita relationships.

Line 605-606. For me, this is one of the headlines of your research. But I don't think it is well enough foregrounded throughout. Make this pop out to readers.
We have included this finding specifically in the abstract and elaborate on it in the discussion.

Line 643-645. Does Alois Kuaso not reach the input threshold for authorship? It seems odd to acknowledge input with no authorship.

Thank you for your concern regarding Alois input. We had offered authorship to him and other contributors. He preferred to be mentioned in the acknowledgements.

Reviewer #3 (Remarks to the Author):

The authors present a manuscript on the recent population history of coastal New Guinea, using ancient DNA from the last 2500 years. This is the first ancient genome data reported from this important but somewhat neglected part of the world, and thus represents a valuable contribution. DNA preservation is presumably not very good in this region, and so the success in generating data is an impressive achievement. The authors then use this data, together with isotope and dental calculus data, to study the population history of the region. The paper provides quite extensive background and archaeological context on the Austronesian and Lapita expansions, and is overall well-written. The analyses are standard and

appear well-executed. I think some clarifications are needed in terms of how analyses, public data, figures/tables etc. are described, as I elaborate on at various points below. I might have expected some more results on within-PNG structure and changes in addition to the focus on the Austronesian expansion and East Asian admixture, but it is of course up to the authors to choose what aspects to focus on.

We thank the reviewer for their kind assessment of this study.

Regarding the integration of the ancient genomes and those of people living in the regions today, we deliberately chose not to link the datasets. In various regions in PNG, including the South Coast, some communities are debating over land rights. In such a climate, an analysis proposing ties to one or another present-day group can have profound and adverse implications, whether intended or not. In this specific case, the two evaluated sites provide only a first and incomplete understanding of the general genetic landscape in the past. Additionally, the obscured and poorly documented historic mobility of groups in the colonial New Guinea caused by colonial episodes of violence adds to the challenge of linking the past to the present without misinterpretation.

I think it would be useful to report in the abstract the number of individuals from which genomic data was recovered – this is useful for readers wanting to get a quick overview.

The abstract was rewritten and this information included

The results obtained on the East Asian admixture question are interesting, but not necessarily very groundbreaking. Most (all?) of the samples substantially post-date the arrival of Austronesians to coastal PNG, so it's not surprising that they do have East Asian admixture or that they have it in variable amounts (just like it's variable today in PNG). The most interesting part of these results is the admixture timing (it would be useful if the sentence "performed an admixture dating analysis" also mentioned what method/approach was used, e.g. at the very least linkage disequilibrium). The result interestingly indicate admixture occurred not until 1,000 years after Austronesian arrival, suggesting a quite long period without admixture – though caveats apply, which the authors discuss appropriately.

Thank you for your comment, we have added the mention of the used tool (DATES) to the manuscript, to spare the broad readership of NatEcoEvo we provide further details only in the supplement.

I could not find any description of what modern datasets were used in this study. This must be described, and source publications cited. For example, there is reference to data from the present-day group Tolai, but it's unclear what this data is – whole-genome data, array data, how many individuals, from what publication? Populations labels like "Denisova_published.DG" are not particularly useful – a few people will know where this data derives from, but this is not sufficient information to enable other researchers to understand what was done and to replicate the analyses.

Thank you for pointing this out. It seems that all citations of datasets used in certain analyses was lost in an editing process. We have reintroduced the relevant citations in the supplement at the beginning of the "Population Genomic analysis section".

The IBD and ROH analyses are somewhat interesting I suppose, but the results are not so easy to interpret when there is no present-day context. It would have been natural to ask how the results in the ancient genomes compare to present ROH levels in PNG, but the authors do not do this (there is quite a bit of modern data from PNG available from previous studies). Though I would not necessarily demand that the authors expand the analysis in this way, as it would be quite a lot of work.

We share your interest in the results. However, coming back to the difficulties of connecting the past to the present in a partly charged political environment, we believe that some results of this type of analysis can reveal quite sensitive information and have negative implications for the present-day communities, i.e. show high rates of consanguinity. We would not like to publish the results without the consent of the original donor-populations and their input to this analysis.

I can't quite get my head around Figure 2 – some samples appear to be missing? I.e. I can't locate any symbols corresponding to present-day PNG, Australia, Bismarck or Solomons. There are text labels for these groups annotated onto the plot, but seemingly no points around those labels? Perhaps a figure processing error.

We apologize for the inconvenience and thank you for pointing this out. There was indeed an error in the plotting, which has now been fixed. The symbols correspond to those given in the legend below the plot. Labels in the plot are intended to give the average non-popgen reader a rough guide when examining the plot.

Supplementary Figure 2 is extremely low-resolution and completely illegible, so the claims made in relation to these results cannot be reviewed currently.

We apologize for this inconvenience and submit a full resolution version with the revised manuscript.

Line 265 onwards describes how two individuals “were consistent with exclusively Papuan-related ancestry”, but line 278 says “Having established that none of the individuals derived from only one stream of ancestry”. These statements do not seem compatible.

Thank you for pointing this out. The sentence now clarifies “Having established that, except for WAT005/B14 and WAT006/B14, none of the individuals derived from only one stream of ancestry using qpWave (Supplementary Table S12) [...]”

There is confusion with regards to the numbering of supplementary tables: the numbers listed in the supplements PDF, the numbers on the Excel file sheets, and the numbers on the first lines of each table, do not agree with each other.

Thank you for pointing this out we will thoroughly check that the numbers do align and apologize for the confusion.

Supplementary Table 12/7 qpWave/qpAdm. It's a bit tricky to understand what the various tables mean. In particular the first table, which has “ID”, “Group”, and “p-value” as column headings – I can't figure out what this table is. Also, more information is needed on the population “New_Guinea” used in the reference community – where precisely in New Guinea this is from is of course highly relevant in this context. Some fields are colour-coded (red, grey), but what this means is not explained. I would recommend against colour-coding as a means to convey information, as this is not machine-readable.

Thank you for pointing that out, we had not considered machine-readability but agree with the importance. The color-code was meant for easier human processing of the tables in regards to significant/non-significant p-values and the reason for the rejection of certain models. We have now specified this in the table caption.

Admixture graph testing for the peopling of the Marianas islands: as there are no suitable ancient genomes from the Philippines, the authors rather leave that possible ancestry unmodelled in the graphs. But should it not be fine to use some present-day Austronesian group from the Philippines, e.g. Kankanaey/Igorot, who seemingly do not have any Papuan-related ancestry (as done in Skoglund et al. 2016). It seems like that would help constrain things, and then hopefully allow a firmer conclusion on this question?

Thank you for this suggestion. We selected the Kankanaey data from the HGDP (Igorot.DG) where complete genomes are available and allow us to use the complete set of 1,24M SNPs we genotyped in all individuals/populations. Kankanaey (Igorot.DG) are genetically very similar to Amis and, when we fit them in our scaffold tree (without WAT002), the best placement is obtained when they are modelled as a sister group of Amis (worst $|Z|=2.8$). Adding those genomes thus only provides information regarding the relationships between Taiwan and the Philippines, and not about the peopling of the Marianas. Notably, a model where the Marianas branch after Amis, but before Igorot.DG is rejected ($|Z|=10.88$). Thus, with the tested data it is not possible to reach more concrete conclusions on the settlement sequence. We have included the trees involving Igorot.DG in the supplementary per your suggestion.

Data availability: The ENA accession number appears inactive, so the archived data cannot be evaluated at this point. It's important that all of the reads, not just those that mapped to the reference genome, are archived (see: <https://www.biorxiv.org/content/10.1101/2023.05.15.540553v1>).

The data is uploaded on ENA but will be published with the paper, as will the genotype on the Poseidon repository. We have uploaded all the sequencing data, adapters removed, but not filtered or aligned.

Response to the reviewers

We thank all reviewers for their time and effort in assessing our study. We have carefully considered the points raised and provide the responses below marked in bold font.

Reviewers' comments:

Reviewer #1 (Remarks to the Author):

I have now completed my second round of review of the manuscript submitted by Nāgele et al. I have no further comments to the authors beyond the initial suggestions/concerns raised after reviewing the first version of this work.

I will, however, express my disappointment in the authors' seeming reluctance to try and correlate their results to linguistic diversity or the genetic history of the region (as suggested by reviewer 3 on multiple instances). In particular, I found their arguments for not performing further analyses, which would clearly strengthen the manuscript, to be very weak. This necessarily leaves me wondering if the reason for their resistance is instead more related to the amount of work (and time) necessary to implement them, or the fact that such analyses are not typical of ancient human DNA studies and thus not as trivial to implement in a standardised/industrial fashion.

I thus find the revised version of the manuscript unsatisfactory, as some of the main comments raised by the reviewers were not properly addressed. The authors did an exceptional job correcting the numerous errors identified in Figures, legends, or nomenclature, but showed an odd reluctance to include additional hypotheses-testing. I still find the manuscript to be solid work, and thus suitable for publication in *Nature Ecology and Evolution*. However, my recommendation will necessarily be to urge the authors to perform the requested analyses before publication.

We appreciate the reviewer's concerns and efforts to improve the manuscript. We want to reassure that the reluctance to compare in a more direct manner to an extended list of present-day groups is not influenced by difficulty of analysis. In fact, imputation and IBD calling is by now part of the majority of aDNA projects. While it is generally (mathematically) unlikely to find IBD links between the small selection of present-day individuals published from the region and the even fewer ancient individuals, in this specific case, the decision to not directly connect the past with the present genetic landscape has ethical and political reasons. We have included our concerns and provide some recommendations on the use of the openly available data in a section in the Supplement "Ethical considerations for the analysis of ancient genomes". In the current climate in PNG, where inter-group violence is a serious concern, a result connecting the (limited) ancient groups with one but not the other present-day group, could aggravate the violence. We therefore strongly feel it would be irresponsible to potentially produce a result that – when published – would be impossible to control, as we have seen that attempts to correct narratives abusing aDNA results are not successful (i.e. <https://www.smithsonianmag.com/history/when-ancient-dna-gets-politicized-180972639/>). We therefore still chose not to make the direct comparison by drawing direct IBD links from the past individuals to those inhabiting the region at present. We have however included the published grouping by region into our formal statistics, investigating access allele sharing of the respective groups and the newly produced ancient individuals compared to New Guinea Highlanders. We have included the results of this analysis in the Supplementary Table S11, and find that there is no group significantly closer to the ancient groups when compared to New Guinean Highlanders.

Reviewer #2 (Remarks to the Author):

Overall, I am highly satisfied with the responses of the author to all of the reviewers' critiques (but see a couple of specific remaining problems below). They have engaged thoughtfully with issues of terminology and how their results are narrated for a broad scientific readership.

There remain some grammatic errors in the manuscript, but I presume Nature will work with the authors on this in the next stage of publication.

For example, in the Abstract (which is much more readable, and conveys better the significance of the research) it says, "The individuals span 2500 years of human habitation and highlight the influences". But no individual spans 2500 years, and no individual highlights anything (rather, their results and interpretation do the highlighting, and their oldest sample is from 2500 years ago). This is a problem of expression, and it is not isolated.

Despite saying they have amended the references for the Hiri and Papuan Hiccup, they now erroneously cite Skelly and David 2017 for the (unproven) idea that there was a breakdown in marine resources caused by climatic changes (Line 522). Skelly and David emphasise social transformations including the possibility that increased warfare led to communities relocating inland to defensive settlements (see Rhoads also!). Where is this mentioned in the manuscript? Only some quite vague climatic evidence is invoked instead.

Thank you for pointing this out. There seems to have been a shift in the citations, as most citations inserted here do in fact not mention changes in the marine resource patterns. We have revised the citations here. We have removed the reference to changes in marine resource patterns. After revision of the literature, we would agree that observations from few sites might not warrant a specific mention of this pattern.

The authors still need to make clear that Rhoads (1982) developed the concept of the Papuan Hiccup & Irwin first developed the Ceramic Hiccup.

We have added citations to the introduction and expanded on the difference of the concepts in the discussion.

They should also briefly mention the various social, economic, etc. explanations that have been put forward for this era beyond climate (Allen's 2010 paper is not the most up to date work on this). The final two chapters of Skelly and David's book will be key for dealing with explanations for the post-650 BP era as it is the most comprehensive review of arguments about this period.

It also seems odd that the authors have introduced a new and unqualified reference to a 'climatically challenging' period immediately prior to 650 BP in the Abstract. Previously the authors more tentatively stated in the main text that an improving climate 'possibly' enabled the post-650 BP transformations. Where are the independent climate records to back this 'challenging period' for PNG's south coast? What do the cores at Lake Kutubu tell us?

I know the authors have limited words to spend, but it is important that they convey that there is no consensus on what drove the post-650 BP re-emergence of coastal settlements and long-distance exchange networks.

We thank Reviewer #2 for their attention to detail and their patience in the description of the archaeological context. Thanks to a slightly increased word limit, we have tried to provide a more detailed description of the Papuan/Ceramic Hiccup, and a more nuanced description how this presents in different

areas on the south coast. We hope that the discussion now also reflects better the significance of the exchange networks for the human activity in the region. The majority of changes can be found in discussion which now reads as follows:

“The archaeological record of the south coast of PNG shows “pulses” of settlement intensification and ceramic abundance (David 2008). This period, coined the “Ceramic Hiccup” (Irwin 1991) but also known as “Papuan Hiccup” (Rhoads 1982) has been suggested to be the result of an abandonment or relocation of sites, with a disruption in ceramic production (Frankel and Rhoads 1994, David 2008) and styles (Rhoads 1982, Bickler 1997), or the contraction of interactions to the eastern parts of the southern PNG coast (Skelly 2014, Skelly and David 2017), where networks were maintained, albeit less intensive (Irwin 1991). This period is believed to have lasted between ~1200 – 500 BP, however the exact timing remains uncertain as a result of insufficient radiocarbon dating (Skelly and David 2017). The assumed period coincides with the medieval climate anomaly (1250 – 700 BP) (Barr, Tibby et al. 2019), where an increase in El Niño southern oscillation events may have led to seasonal droughts, suggesting the changes in settlement patterns could have been, at least partly, influenced by the availability of fresh water and the viability of cultivation (Allen 2010, Shaw, Coxe et al. 2020). Whether or not climate had a direct effect on the coastal communities, the interruption of long distance trade links might have led to abandonment of many long term settlement locales, relocation further inland (Rhoads 1982), and perhaps also been catalyst for conflict between groups. Certainly, settlement of Nebira 4, at the base of the Nebira hill, was abandoned when settlement of Nebira 2, on the saddle of the double-peaked hill commenced (Allen 1972, Bulmer 1975, David, Aplin et al. 2015). A shift to defensive hilltop settlement suggests that the relationships between groups might have changed, and defensive sites on hilltops became the preferred occupational area (Skelly and David 2017).

The majority of individuals from Nebira 2 show an admixture event between 1600-1100 BP, overlapping with this period. The individual NBR020/ACJ-34 shows an admixture date of only ~650 years, much younger than the majority of the group. Additionally, this individual displayed a non-local $^{87}\text{Sr}/^{86}\text{Sr}$ ratio (Shaw, Buckley et al. 2011), and a different childhood diet, suggesting they may have originated from a population with a different ancestral history. The admixture date of NBR020/ACJ-34 coincides with a period when coastal and inland settlements had become archaeologically visible, and ceramic wares show stylistic innovations with regional variation indicating the establishment or reinforcement of social boundaries (Vanderwal 1973, Allen 1977, Allen 1977, Bulmer 1982). Trade supposedly resumed, perhaps with reconfigured networks, in ~800-500 BP, and has been associated with different Austronesian-speaking groups arriving on the coasts (Bergström, Oppenheimer et al. 2017).

Overlap of maritime trade networks such as the Kula and *hiri* trade networks by at least 500 BP (David 2008, Skelly and David 2017, Irwin, Shaw et al. 2019, Liu, Peter et al. 2022), suggests a (re-) emergence of these networks in the period following the “Papuan Hiccup” (Skelly and David 2017).

Admixture dates inferred from the individuals from Eriama (Fig. 3d), together with the higher Papuan ancestry (Fig. 3a, e) indicate a continuation of genetic exchange beyond that identified for Nebira 2. The strong shift from 40% to 15% Asian-related ancestry suggests they were part of an interaction sphere with people carrying higher Papuan-related ancestry, in the model represented by Papuan Highlanders. However, we lack the resolution to identify which populations contributed and where their geographical source was. During the “Papuan Hiccup”, communities might have retreated into the interior areas, where interactions with highland populations might have intensified (Rhoads 1982). One individual from Eriama (ERI007/ACV-7) shows more recent admixture inferred to around 600-400 years ago, also implying that the population in Eriama continued genetic exchange with other populations.

Despite the many differences between the two sites, analyses of IBD blocks reveal distant connections between Nebira 2 and Eriama, suggesting that they constituted one reproductive unit until ~13 generations before the analysed population, estimating the split to ~ 640 BP. The different genetic compositions of the two nearby sites show that the southern coast was a genetic, and possibly also a cultural and linguistic mosaic of people, matching the situation in Motu and Koita speaker communities today. The Motu language is a Central Papuan Tip language, of the Western Oceanic branch of the Oceanic group of

Austronesian languages (Blust 1990, Ross 1994), and is spoken mostly by people living in the coastal region of the Central Province. The Papuan language dominant in the region, Koita (Dutton 2010), is spoken in settlements more inland (Dutton 1969, Swadling 1977, Oram 1981, Swadling 1981). The two major cultural groups, occupying the coastal and interior areas in the vicinity of Nebira 2 and Eriama, have historically close social connections that regularly included intermarriage, with both groups involved in the maritime *hiri* trade expeditions (Seligman, Barton et al. 1910).

The oral traditions of present-day groups remembering their ancestors residing at Nebira 2 and Eriama include tales of a relocation from the mountains (Dutton 1969, Bulmer 1978). The reason for the separation of the two communities, despite no evident geographical barriers, remains enigmatic, and might point to the introduction of a cultural barrier connected to different interactions spheres of the two groups after relocation. “

Reviewer #3 (Remarks to the Author):

The authors have responded well to the reviewer comments, and clarified various things in figures and tables. I have no further comments, and would recommend the paper for publication. I find the Discussion a bit overly long personally, but it is up to the authors to decide. I just leave a couple of final minor things below.

Figure 3a: It needs to be indicated, at least in the figure legend, what the numbers on the right hand-side are (presumably these are p-values). Also, the numbers currently overlap the figure. It also needs to be indicated what the error bars represent.

“an excess of Austronesian-related ancestry on the X-chromosome ranging from 10 to 60 percent” – should this rather say “percentage points”, instead of “percent”?

Supplementary Figure S2 is very low-resolution, it’s virtually impossible to read the population labels (they also seem to be upside down?).

We thank Reviewer #3 for their kind assessment and the continued effort to spot details that can be improved. We have added the description of the p-values into the caption of Figure 3, and followed the suggested wording in the description of the sex-biased ancestry.

we apologize for the resolution issue and have attempted to further increase resolution, but will also be in contact with the publication team to make sure a full resolution picture is available separately.

Remarks from the previous Revision stage:

Reviewer #1 (Remarks to the Author):

Nägele et al. present a new ancient DNA study of the genetic ancestry of Papua New Guinea and the Bismarck archipelago, including new archaeological information (radiocarbon dating and isotope analysis).

The authors sought to establish a time-transect that extends close to the initial settlement of the Lapita

cultural complex in Near Oceania, employed commonly used tools in ancient DNA ancestry modelling (e.g., PCA, f-statistics, qpAdm, IBD block analysis) and a design strategy that reflects the extensive experience of this group of researchers doing very similar analysis in the past. Accordingly, I do not have any major comments on the strategy or the methods used in this work - which are standard and frequently replicated - only a few comments regarding the results.

1. Given the geographical proximity and synchronous sampling of the Nebira and Eriama sites, I share the authors surprise (lines 557-560) that the inferred history of the two populations, in particular the proportion of New Guinea ancestry, is so disparate.

If this is the case, then the most reasonable explanation, also suggested by the authors, is for the existence of a strong cultural barrier causing the observed population structure. In New Guinea, the most obvious cultural barrier would be language. The authors seem to think so as well (lines 561-571), but I found the sole paragraph on this possibility unsatisfactory. I thus think the manuscript would benefit from a more detailed comparison of the history of language evolution in this specific region, beyond the sparse information provided. Is the language barrier hypothesis testable? There is extensive literature on the languages of PNG that could perhaps be leveraged to test this hypothesis (or at least strengthen it). Given the multidisciplinary approach to this study, I think this could be an important addition.

We thank the reviewer for their agreement. As we discuss in the manuscript, language can be cultural barrier between communities, however, in situations where many languages exist side-by-side in a small area, lots of language mixing can be observed. This is particularly the case in the Western Pacific and generally in New Guinea. This observation is a product of cultural mixing and intense/prolonged social interaction both so much so historically that they create obstacles for historical linguists in classifying some languages. With this regard, we chose to not go into depth discussing or testing this in our manuscript, as the historical impacts would lead to possibly wrong conclusions when putting together with the ancient DNA data. Linguistic diversity could be a cultural barrier in these particular areas of New Guinea, and it would be in principle possible to test this, however this would be a much larger separate study.

However, could this observation instead be the result of an analytical artefact? I think it unlikely. But could the authors provide some evidence to rule this possibility out? For example, the aDNA analysis is based on captured data - how many 1240k sites have been included per individual, and do they overlap between the populations? Can you restrict the analysis to overlapping sites (even if needing to exclude low-coverage individuals and despite losing some power)? Such information would be more reassuring of the observations and rule out any potential ascertainment bias.

The individual SNP coverage for the two sites is (for tropical environments) surprisingly good, with an average coverage of >430 000 sites for Nebira and > 640 000 for Eriama, ranging between 11 000 to 610 000 and 430 000 to 790 000 respectively (see table S1 for an overview). The number of SNPs overlapping the two should be >103 000 SNPs, as indicated as the overlap used to calculate the f4 statistic Mbuti.DG, Papuan.DG; Eriama, Nebira, added to the Supplementary data Table S11 (Papuan related ancestry section column I, line 5). This test has a strongly negative result, showing the excess affinity of Eriama to Papuan, when compared to Nebira. We would therefore argue that a bias based on the coverage can be excluded, and we do observe a true signal, consistent with qpAdm, Admixture and PCA.

2. Figure 1: color codes are not clear. I assume they represent different populations, but cannot be sure. Please include an explanation in the figure legend.

The different colours represent the different archaeological sites/complexes and are used to represent individuals from these sites in the PCA. We have clarified this by adding the statement to the figure caption.

3. Figure 2: Again, I am assuming that the color codes represent different populations. It is a bit challenging to examine the PCA plot, but it seems to me that there are some discrepancies regarding the number of

individuals projected in the PCA and the numbers given in Figure 1. Specifically, there are 9 blue circles projected for Eriama (if I am observing correctly), but the authors refer only 8 individuals were analysed in Figure 1. In addition, I could not find the two Tilu individuals in the PCA. (The numbers for Nebira and Watom do seem correct).

Thank you for pointing this out. The mistake in the plotting was corrected. Tilu is now shown as a square, TIL004 is transparent because of the low number of SNPs, as are NBR008 and WAT006.

Also, the statement in lines 255-258 should be rephrased. The fact that individuals cluster between two population groups (i.e., East Asians and Papuans) in the PCA does not mean they have 50% ancestry from each group, as it is now stated. In fact, the authors later show (qpAdm) that these proportions vary between the two populations.

We agree with the reviewer that the placement does not reflect a 50% percent ancestry from each group, and realize that the phrase, although not intended to state that, did suggest so. We have now changed the wording to state the intended conclusion that “*The placement of the Eriama and Nebira individuals suggests a genetic composition derived from a mixture of Papuan-related ancestry most similar to present-day highland populations of New Guinea (Supplementary Fig. 2) and of East Asian-related ancestryA.*”

4. Figure 3d: There seems to be a circle overlapping the filled block representing the admixture timing range for individual ERI007. Also, individuals ERI003, ERI006 and ERI009; and individuals ERI004 and ERI005 seem to have the exact same AMS dates (including range) - please double check.

Thank you for pointing out another plotting artifact, the overlapping dot has been removed. For a maximum conservative approach, we show the date ranges of the calibrated dates BP 95% probability range, as reported in extended data Table S2. Despite slightly different uncalibrated dates, the calibration does equalize the dates reported for the respective individuals. For increased transparency, we have added the details on the reported dates to the Figure 3d caption.

5. Line 458: replace "Firgure 2" by "Figure 2". **Changed.**

6. Line 502: where is Figure 3e? I think the authors mean 3a. **Correct, thank you for pointing that out.**

Reviewer #2 (Remarks to the Author):

This paper details the implications of the first ever ancient DNA analyses from Papua New Guinea, a key location for understanding ancient migrations (c. 65,000 years ago into Sahul) and a staging post for later Oceanic migrations (of Austronesian peoples after c. 3500 years ago). The samples size is adequate, given the rarity of aDNA analyses in Oceania, though as the authors note only two samples have any bearing on the question of when ‘East Asian’ and ‘Papuan’ DNA admixed after Austronesian arrival in coastal New Guinea. Most of their samples are from very recent sites (the past 500 years) but nevertheless tell us something about admixture through time. Probably the study's most significant findings is that "the first, minimally admixed settlers remained isolated from local populations with Papuan-related ancestry, despite geographical proximity". However, the term "isolated" here is very problematic - they may have shared close relationships (e.g., exchange) but not intermarried. Which is very interesting indeed.

We thank the reviewer for their caution regarding terminology used and agree – “isolation” is more than just the absence of genetic mixing. To add nuance, we have changed the wording to “genetic isolation”.

The data and supplementary information are of a high quality, and there are no obvious errors. I must caution that aDNA is not my field, so my comments are on the narrative and the archaeology (including the radiocarbon dating).

I am slightly concerned about the vagueness of the term East Asian – I can tell from the Supplementary Information that the authors have given thought to terminology. Is there an alternative that cannot be confused with modern sociopolitical regions? Could “Indigenous East Asian” be used, given Indigenous Taiwanese aDNA samples were used as a comparison? Or Austronesian? (This has its own problems, I am aware).

On a related note, the relationship between Austronesian dispersals and East Asian ancestry needs to be mentioned in the Abstract and early in the Introduction.

Thank you for regarding the terminology. We have also given thought to the term for the broad genetic signal observed, and decided here to use a broader assignment of “East Asian”. Considering “Indigenous East Asian” when mostly Taiwanese aboriginal genomes were used for the modeling, disregards other indigenous East Asian groups. At the same time, we wanted to avoid using “aboriginal Taiwanese” as a term for a genomic signal preceding the naming of the island as Taiwan (relating to your own concerns of present-day sociopolitical regions). Considering the use of “Austronesian” for the genetic signal, we also decided to avoid this language-based naming because Austronesian speaking groups today are very variable in their genetic ancestry, the cultural representation and geographic extent of the language. Similar to the oversimplification of the diverse Papuan ancestry profiles to “Papuan-related”, we decided to default to a broad term of “East Asian-related”, not inspired by sociopolitical regions, but by the geographical region that is most likely the ancestral home of the genetic profile dispersing in East-, South East- and Island South East Asia ~7000 – 5000 years ago. We have added a paragraph to the “Terminology” section of the Supplement to explain the terminology:

“A similar problem arises when describing the general genetic profile deriving from a dispersal from Asia. In this study, we refer to the genetic component associated with a dispersal from the mainland and island South East Asia as “East Asian-related”, similar to the oversimplification of the diverse Papuan ancestry profiles to “Papuan-related. This term should not be understood as a reflection of a sociopolitical region, but inspired by the broad geographical region that is most likely the ancestral home of the genetic profile dispersing in East-, South East- and Island South East Asia ~ 7000 -500 years ago. Alternative terms could be “Indigenous East Asian”. However, since mostly Taiwanese and Philippine indigenous genomes were used for or the modeling this would disregard the many indigenous East Asian groups with different genetic profiles. Considering the use of “Austronesian” for the genetic signal, we also decided to avoid this language-based naming because Austronesian speaking groups today are very variable in their genetic ancestry, the cultural representation and geographic extent of the language.”

In my opinion, the Title and Abstract need rewriting to show readers the headline of your findings. “The impact” is a bit vague. The Abstract too needs to convey your main findings in more plain language, and there is currently a very awkward transition between the 2700-2100 results and the ‘Three coastal sites’ which sounds like an entirely new Abstract/paper is starting on line 69. This needs rewriting to flow better.

Thank you for this suggestion. We have rewritten the abstract and hope that it now conveys the main findings more efficiently regardless of the limit of 200 words imposed by the journal.

One area the paper needs sharpening is in its discussion of the Papuan Hiccup. Given that most of the samples are from the past 500 years, and admixture is inferred not long before that, much of the actual data relates to a period just prior to the emergence of vast ethnographically known exchange routes (e.g., Hiri). However, the authors have incorrectly cite Shaw and Allen for this period, when first mention was made

by Geoff Irwin and Jim Rhoads – since this time the most extensive discussion of the period is in Skelly and David, and only Jim Allen attributes the abandonment of coastal sites to “challenging environmental fluctuations” – the authors should expand their discussion upon reading the literature on this period more closely (add nuance). It is not surprising that there is a link between the genetics, the development of the hiri, and the development of Motu-Koita interrelations, and this could perhaps be explained more clearly also.

Thank you for suggesting additional literature. We have carefully considered the suggested readings and hope to have included them to your satisfaction, some of those edits detailed in the suggestions below.

Small notes:

Line 56. What are ancient inhabitants? Why not just inhabitants?

The term ancient inhabitants was used to contrast the present-day inhabitants, whose genomic information is available to some extent. However, we see how this wording could be understood as excluding people today from an impact in the region, and have therefore removed “ancient” from this sentence.

Line 83. And the Aru Islands? **Added to the sentence**

Line 142. Cite original papers for Papuan/Ceramic Hiccup

We thank the reviewer again for the suggested literature and have now updated the citation and expanded the explanation of the phenomenon.

Line 152. We need to get a sense of your argument in this part of the introduction

Line 170. Should be ‘Affect how’

Line 354. What are ‘seafaring probabilities’?

We thank you for pointing out the use of jargon here and have replaced the term with “highest probabilities of landfall in the Marianas during a seavoyage”

Line 375. Why is Bulmer not cited?

Is now cited, as should have been.

Line 383. This is a major issue for me and needs fixing. Of course ‘family was not necessarily restricted to genetic relatedness’ – any familiarity with the Indigenous societal structures of PNG would make this very clear, and some nuance needs to be shown here. Western genetics is not a holistic analysis of ‘family’ or ‘kin’ – these are socially constructed.

Thank you for this comment. We completely agree with the notion that genetic cannot singlehandedly provide insight into family and kin. However, combined with other lines of evidence, such as consumption patterns, distribution of burial goods and burial customs, it can help interpreting the archaeological context. In the current study, unfortunately none of the patterns observed would allow us to delve into such questions and compare the patterns with those observed in present-day communities where we know how kin and family are defined. The statement that “family is not necessarily restricted to genetic relatedness” is addressed at a readership that might not be familiar with the breadth of definitions of kin in various cultures and parts of the world.

Line 505. This inferred date is inferred, and probably due to your sample size. Suggest re-emphasising the limited dataset. It is highly improbable that admixture took more than 1000 years

We do share your skepticism regarding the significant delay between the arrival of Lapita pottery makers and the detection of the genetic admixture. From the genetic line of evidence we are currently unable to look past the admixture that occurred 1000 years ago. Hopefully the investigation of a time transect in the region will help to improve the inference (exemplified in Posth, Nägele et al. 2018). Until then, we will have to report the date of 1000 BP, of course pointing to the limitations, which we hope we were able to convey better in this version, regardless of the significant cuts to meet the word limit.

Line 519-521. This alternative explanation is completely unsupported by the archaeological evidence. Despite a great deal of debate back in 2012 about whether Lapita people ever visited any mainland, the entire Lapita community accepts there were settlements – if they had only found pottery, this would be a reasonable statement.

Thank you for raising this point, we agree and have removed this statement.

Line 539. Rhoads and Urwin need citing for hiri emergence - Shaw et al for the Kula.
We have updated the citations.

Line 550. Reflect here on the emergence of Motu-Koita relationships/kinship.
We have included a statement relating the observed pattern to the Motu-Koita relationships.

Line 605-606. For me, this is one of the headlines of your research. But I don't think it is well enough foregrounded throughout. Make this pop out to readers.
We have included this finding specifically in the abstract and elaborate on it in the discussion.

Line 643-645. Does Alois Kuaso not reach the input threshold for authorship? It seems odd to acknowledge input with no authorship.
Thank you for your concern regarding Alois input. We had offered authorship to him and other contributors. He preferred to be mentioned in the acknowledgements.

Reviewer #3 (Remarks to the Author):

The authors present a manuscript on the recent population history of coastal New Guinea, using ancient DNA from the last 2500 years. This is the first ancient genome data reported from this important but somewhat neglected part of the world, and thus represents a valuable contribution. DNA preservation is presumably not very good in this region, and so the success in generating data is an impressive achievement. The authors then use this data, together with isotope and dental calculus data, to study the population history of the region. The paper provides quite extensive background and archaeological context on the Austronesian and Lapita expansions, and is overall well-written. The analyses are standard and appear well-executed. I think some clarifications are needed in terms of how analyses, public data, figures/tables etc. are described, as I elaborate on at various points below. I might have expected some more results on within-PNG structure and changes in addition to the focus on the Austronesian expansion and East Asian admixture, but it is of course up to the authors to choose what aspects to focus on.

We thank the reviewer for their kind assessment of this study.

Regarding the integration of the ancient genomes and those of people living in the regions today, we deliberately chose not to link the datasets. In various regions in PNG, including the South Coast, some communities are debating over land rights. In such a climate, an analysis proposing ties to one or another present-day group can have profound and adverse implications, whether intended or not. In this specific case, the two evaluated sites provide only a first and incomplete understanding of the general genetic landscape in the past. Additionally, the obscured and poorly documented historic mobility of groups in the

colonial New Guinea caused by colonial episodes of violence adds to the challenge of linking the past to the present without misinterpretation.

I think it would be useful to report in the abstract the number of individuals from which genomic data was recovered – this is useful for readers wanting to get a quick overview.

The abstract was rewritten and this information included

The results obtained on the East Asian admixture question are interesting, but not necessarily very ground-breaking. Most (all?) of the samples substantially post-date the arrival of Austronesians to coastal PNG, so it's not surprising that they do have East Asian admixture or that they have it in variable amounts (just like it's variable today in PNG). The most interesting part of these results is the admixture timing (it would be useful if the sentence "performed an admixture dating analysis" also mentioned what method/approach was used, e.g. at the very least linkage disequilibrium). The results interestingly indicate admixture occurred not until 1,000 years after Austronesian arrival, suggesting a quite long period without admixture – though caveats apply, which the authors discuss appropriately.

Thank you for your comment, we have added the mention of the used tool (DATES) to the manuscript, to spare the broad readership of NatEcoEvo we provide further details only in the supplement.

I could not find any description of what modern datasets were used in this study. This must be described, and source publications cited. For example, there is reference to data from the present-day group Tolai, but it's unclear what this data is – whole-genome data, array data, how many individuals, from what publication? Population labels like "Denisova_published.DG" are not particularly useful – a few people will know where this data derives from, but this is not sufficient information to enable other researchers to understand what was done and to replicate the analyses.

Thank you for pointing this out. It seems that all citations of datasets used in certain analyses was lost in an editing process. We have reintroduced the relevant citations in the supplement at the beginning of the "Population Genomic analysis section".

The IBD and ROH analyses are somewhat interesting I suppose, but the results are not so easy to interpret when there is no present-day context. It would have been natural to ask how the results in the ancient genomes compare to present ROH levels in PNG, but the authors do not do this (there is quite a bit of modern data from PNG available from previous studies). Though I would not necessarily demand that the authors expand the analysis in this way, as it would be quite a lot of work.

We share your interest in the results. However, coming back to the difficulties of connecting the past to the present in a partly charged political environment, we believe that some results of this type of analysis can reveal quite sensitive information and have negative implications for the present-day communities, i.e. show high rates of consanguinity. We would not like to publish the results without the consent of the original donor-populations and their input to this analysis.

I can't quite get my head around Figure 2 – some samples appear to be missing? I.e. I can't locate any symbols corresponding to present-day PNG, Australia, Bismarck or Solomons. There are text labels for these groups annotated onto the plot, but seemingly no points around those labels? Perhaps a figure processing error.

We apologize for the inconvenience and thank you for pointing this out. There was indeed an error in the plotting, which has now been fixed. The symbols correspond to those given in the legend below the plot. Labels in the plot are intended to give the average non-popgen reader a rough guide when examining the plot.

Supplementary Figure 2 is extremely low-resolution and completely illegible, so the claims made in relation to these results cannot be reviewed currently.

We apologize for this inconvenience and submit a full resolution version with the revised manuscript.

Line 265 onwards describes how two individuals “were consistent with exclusively Papuan-related ancestry”, but line 278 says “Having established that none of the individuals derived from only one stream of ancestry”. These statements do not seem compatible.

Thank you for pointing this out. The sentence now clarifies “Having established that, except for WAT005/B14 and WAT006/B14, none of the individuals derived from only one stream of ancestry using qpWave (Supplementary Table S12) [...]”

There is confusion with regards to the numbering of supplementary tables: the numbers listed in the supplements PDF, the numbers on the Excel file sheets, and the numbers on the first lines of each table, do not agree with each other.

Thank you for pointing this out we will thoroughly check that the numbers do align and apologize for the confusion.

Supplementary Table 12/7 qpWave/qpAdm. It’s a bit tricky to understand what the various tables mean. In particular the first table, which has “ID”, “Group”, and “p-value” as column headings – I can’t figure out what this table is. Also, more information is needed on the population “New_Guinea” used in the reference community – where precisely in New Guinea this is from is of course highly relevant in this context. Some fields are colour-coded (red, grey), but what this means is not explained. I would recommend against colour-coding as a means to convey information, as this is not machine-readable.

Thank you for pointing that out, we had not considered machine-readability but agree with the importance. The color-code was meant for easier human processing of the tables in regards to significant/non-significant p-values and the reason for the rejection of certain models. We have now specified this in the table caption.

Admixture graph testing for the peopling of the Marianas islands: as there are no suitable ancient genomes from the Philippines, the authors rather leave that possible ancestry unmodelled in the graphs. But should it not be fine to use some present-day Austronesian group from the Philippines, e.g. Kankanaey/Igorot, who seemingly do not have any Papuan-related ancestry (as done in Skoglund et al. 2016). It seems like that would help constrain things, and then hopefully allow a firmer conclusion on this question?

Thank you for this suggestion. We selected the Kankanaey data from the HGDP (Igorot.DG) where complete genomes are available and allow us to use the complete set of 1,24M SNPs we genotyped in all individuals/populations. Kankanaey (Igorot.DG) are genetically very similar to Amis and, when we fit them in our scaffold tree (without WAT002), the best placement is obtained when they are modelled as a sister group of Amis (worst $|Z|=2.8$). Adding those genomes thus only provides information regarding the relationships between Taiwan and the Philippines, and not about the peopling of the Marianas. Notably, a model where the Marianas branch after Amis, but before Igorot.DG is rejected ($|Z|=10.88$). Thus, with the tested data it is not possible to reach more concrete conclusions on the settlement sequence. We have included the trees involving Igorot.DG in the supplementary per your suggestion.

Data availability: The ENA accession number appears inactive, so the archived data cannot be evaluated at this point. It’s important that all of the reads, not just those that mapped to the reference genome, are archived (see: <https://www.biorxiv.org/content/10.1101/2023.05.15.540553v1>).

The data is uploaded on ENA but will be published with the paper, as will the genotype on the Poseidon repository. We have uploaded all the sequencing data, adapters removed, but not filtered or aligned.

Allen, J. (1972). "Nebira 4: an early Austronesian site in central Papua." Archaeology & Physical Anthropology in Oceania 7(2): 92-124.

Allen, J. (1977). "Management of resources in prehistoric coastal Papua." The Melanesian Environment: 35-44.

- Allen, J. (1977). "Sea traffic, trade and expanding horizons." Sunda and Sahul: Prehistoric Studies in Southeast Asia, Melanesia and Australia: 387-417.
- Allen, J. (2010). "Revisiting Papuan ceramic sequence changes: another look at old data." Artefact: the Journal of the Archaeological and Anthropological Society of Victoria, The(33): 4.
- Barr, C., J. Tibby, M. Leng, J. Tyler, A. Henderson, J. Overpeck, G. Simpson, J. Cole, S. Phipps and J. Marshall (2019). "Holocene el Niño–southern Oscillation variability reflected in subtropical Australian precipitation." Scientific Reports **9**(1): 1627.
- Bergström, A., S. J. Oppenheimer, A. J. Mentzer, K. Auckland, K. Robson, R. Attenborough, M. P. Alpers, G. Koki, W. Pomat and P. Siba (2017). "A Neolithic expansion, but strong genetic structure, in the independent history of New Guinea." Science **357**(6356): 1160-1163.
- Bickler, S. H. (1997). "Early pottery exchange along the south coast of Papua New Guinea." Archaeology in Oceania **32**(2): 151-162.
- Blust, R. (1990). "Three recurrent changes in Oceanic languages." Pacific island languages: Essays in honour of GB Milner: 7-28.
- Bulmer, S. (1975). "Settlement and economy in prehistoric Papua New Guinea: a review of the archeological evidence." Journal de la Société des Océanistes **31**(46): 7-75.
- Bulmer, S. (1978). Prehistoric culture change in the Port Moresby region, University of Papua New Guinea.
- Bulmer, S. (1982). "West of Bootless Inlet: Archaeological evidence for prehistoric trade in the Port Moresby area and the origins of the Hiri." The Hiri in History: Further Aspects of Long Distance Motu Trade in Central Papua **11**: 7-130.
- David, B. (2008). "Rethinking cultural chronologies and past landscape engagement in the Kopi region, Gulf Province, Papua New Guinea." The Holocene **18**(3): 463-479.
- David, B., K. Aplin, F. Petchey, R. Skelly, J. Mialanes, H. Jones-Amin, J. Stanistic, B. Barker and L. Lamb (2015). "Kumukumu 1, a hilltop site in the Aird Hills: Implications for occupational trends and dynamics in the Kikori River delta, south coast of Papua New Guinea." Quaternary International **385**: 7-26.
- Dutton, T. E. (1969). The peopling of Central Papua: Some preliminary observations, Pacific Linguistics, Research School of Pacific and Asian Studies, The
- Dutton, T. E. (2010). Reconstructing Proto Koirian: The history of a Papuan language family, Pacific Linguistics, Research School of Pacific and Asian Studies, The
- Frankel, D. and J. W. Rhoads (1994). Archaeology of a coastal exchange system: sites and ceramics of the Papuan Gulf, Division of Archaeology and Natural History, Research School of Pacific and
- Irwin, G. (1991). "Themes in the prehistory of coastal Papua and the Massim." Man and a Half: Essays in Pacific anthropology and ethnobiology in honour of Ralph Bulmer: 503-510.
- Irwin, G., B. Shaw and A. Mcalister (2019). "The origins of the Kula Ring: Archaeological and maritime perspectives from the southern Massim and Mailu areas of Papua New Guinea." Archaeology in Oceania **54**(1): 1-16.
- Liu, D., B. M. Peter, W. Schiefenhövel, M. Kayser and M. Stoneking (2022). "Assessing human genome-wide variation in the Massim region of Papua New Guinea and implications for the Kula trading tradition." bioRxiv.
- Oram, N. (1981). "The history of the Motu-speaking and Koitabu-speaking peoples according to their own traditions." Oral tradition in Melanesia: 207-229.

Rhoads, J. W. (1982). "Prehistoric Papuan exchange systems: the hiri and its antecedents." The Hiri in History: Further aspects of long distance Motu trade in Central Papua: 131-151.

Ross, M. D. (1994). "Central Papuan culture history: some lexical evidence." Austronesian terminologies: Continuity and change: 389-479.

Seligman, C. G., F. R. Barton and E. Giblin (1910). The Melanesians of British New Guinea, Cambridge, U. P.

Shaw, B., H. Buckley, G. Summerhayes, C. Stirling and M. Reid (2011). "Prehistoric migration at Nebira, South Coast of Papua New Guinea: New insights into interaction using isotope and trace element concentration analyses." Journal of Anthropological Archaeology **30**(3): 344-358.

Shaw, B., S. Coxe, V. Kewibu, J. Haro, E. Hull and S. Hawkins (2020). "2500-year cultural sequence in the Massim region of eastern Papua New Guinea reflects adaptive strategies to small islands and changing climate regimes since Lapita settlement." The Holocene: 0959683620908641.

Skelly, R. J. (2014). From Lapita to the Hiri: Archaeology of the Kouri Lowlands, Gulf of Papua, Papua New Guinea, Monash University.

Skelly, R. J. and B. David (2017). Hiri: Archaeology of long-distance maritime trade along the south coast of Papua New Guinea, University of Hawai'i Press.

Swadling, P. (1977). "A review of the traditional and archaeological evidence for early Motu, Koita and Koiari settlement along the central south Papuan coast." Oral History **5**(2): 37-43.

Swadling, P. (1981). "The settlement history of the Motu and Koita speaking people of the Central Province, Papua New Guinea." Oral Tradition in Melanesia: 240-251.

Vanderwal, R. L. (1973). Prehistoric studies in central coastal Papua, Australian National University.

Response to Reviewer #2:

Remarks to the Author:

My remaining concerns (2nd round of reviews) were about the quality of the archaeological literature review. If the paper had been published in its previous state, scholars working on the south coast of PNG would have read this paper and thought (possibly correctly) that the authors had not actually read the literature describing the emergence of exchange systems (especially the literature from the Papuan Gulf).

The difference between that manuscript and this one is night and day. The authors have produced a highly nuanced and evidence-based discussion that reflects the current state of research and positions their own contribution much more effectively.

I suggest that the authors add a small caveat next to their citation of the Medieval Warm Period, simply stating that the IPCC (<https://www.ipcc.ch/>) is not convinced there is enough evidence for a global (southern hemisphere) climatic era. The tree ring data from Australia actually contradicts this period. There is some positive evidence from Pacific coral (<https://www.science.org/doi/10.1126/science.1240837>) but the authors should very briefly flag that more data is needed (i.e., their paper is important for highlighting the need for future research on the social and climatic drivers for exchange system transformations, etc).

Excellent rewrite, still an unfortunate case of asking external reviewers rather than co-authors to do a fair bit of literature review legwork

We thank reviewer #2 for the continued effort to improve the manuscript. We have taken the final suggestion in consideration, and have added detail to the sentence discussing potential environmental causes for the observed patterns on the South Coast. the sentence now reads as follows:

“The assumed period coincides with the medieval climate anomaly (1250 – 700 BP), which was possibly delayed in Oceania, however a lack of proxy sites from PNG and Western Oceania leaves the magnitude of the effect open for the specific region (Lüning, Gałka et al. 2020). Still, an increase in El Niño southern oscillation events (Lawman, Quinn et al. 2020) may have led to seasonal droughts, suggesting the changes in settlement patterns could have been, at least partly, influenced by the availability of fresh water and the viability of cultivation (Allen 2010, Shaw, Coxe et al. 2020)“.

We have opted to not include a citation to the IPCC, because this information is not easily accessible. Furthermore, the report from 2021 states that "The terms 'Little Ice Age' and 'Medieval Warm Period' (or 'Medieval Climate Anomaly') are not used extensively in this report because the timing of these episodes is not well defined and varies regionally. Since AR5, new proxy records have improved climate reconstructions at decadal scale across the last millennium. Therefore, the dates of events within these two roughly defined periods are stated explicitly when possible." (Page 295 of the 2021 "Climate Change - the physical science basis" document). We hope that reviewer # 2 will agree with our approach to instead point out the poor regional resolution specifically for Western Oceania/PNG, but cite the review of proxy records for the region, suggesting increased temperature also in the Pacific, and the possibly delayed timing of the effect.